# SHED LIGHT ON SEGMENTATION FOR MONOCULAR DEPTH ESTIMATION

## ABSTRACT

Depth estimation is a dense prediction task that infers per-pixel depth from a single image, fundamental to 3D perception and robotics. Although real-world scenes exhibit strong structure, these methods treat it as an independent pixel-wise regression problem, often resulting in structural inconsistencies in depth maps, such as ambiguous object shapes. We propose SHED, a novel encoder-decoder architecture that enforces geometric prior explicitly from spatio-layout by incorporating segmentation into depth estimation. Inspired by the bidirectional hierarchical reasoning in human perception, SHED redesigns the vision transformer by replacing fixed patch tokens with segment tokens, which are hierarchically pooled in the encoder and unpooled in the decoder to reverse the hierarchy. The model is supervised only at the final output, and the intermediate segment hierarchy emerges naturally without explicit supervision. SHED offers three key advantages. First, it improves depth boundaries and segment coherence, and demonstrates robust cross-domain generalization. Second, it enables features and segments to better capture global scene layout. Third, it enhances 3D reconstruction and reveals part structures that conventional pixel-wise methods fail to capture.

## 1 INTRODUCTION

Images are 2D projections of the 3D world, where surfaces, regions, and boundaries form a coherent structure. Many vision tasks aim to recover this structure by predicting semantic or geometric values at each pixel, a process known as dense prediction (Forsyth & Ponce, 2002). Among them, monocular depth estimation is one of the most studied, inferring depth from a single RGB image (Torralba & Oliva, 2002). Despite the inherent structure of real-world scenes, most models, including the Dense Prediction Transformer (DPT) (Ranftl et al., 2021), treat the task as independent pixel-wise regression. Although their outputs may appear plausible, they often lack structural consistency, resulting in ambiguous object shapes (Figure 1, row 1).

This limitation stems from a disconnect between depth estimation and scene organization. Depth encodes geometric structure, while segmentation captures semantically coherent regions. Though serving different purposes, the two are closely related: segment boundaries align with depth discontinuities, and depth gradients with semantic boundaries. This relationship has long been recognized in classical vision literature (Malik et al., 2016), yet recent models such as Depth Anything (Yang et al., 2024b) and Segment Anything (Ravi et al., 2025) treat them as independent tasks, largely overlooking their connection.

In contrast, the human visual system integrates depth and segmentation through a bidirectional hierarchical process (Hochstein & Ahissar, 2002), where part-whole segmentation informs depth estimation, and depth in turn guides segmentation. It first infers a global layout by grouping segments from fine to coarse, then refines depth from coarse to fine, adding detail within smaller regions while preserving the overall structure. This organization supports part-whole reasoning and yields depth maps with sharp boundaries and smooth intra-object variations (Figure 1, row 2).

To realize this idea, we propose a novel architecture called SHED, which performs monocular depth estimation using a bidirectional segment hierarchy. With the design of DPT (Ranftl et al., 2021), a standard encoder-decoder framework built on the Vision Transformer (ViT) (Dosovitskiy, 2020), but replaces fixed-size patch tokens with hierarchical segment tokens to produce a structured depth. These tokens are organized from fine to coarse and learned in an *unsupervised* manner, guided solely by pixel-wise regression objectives.

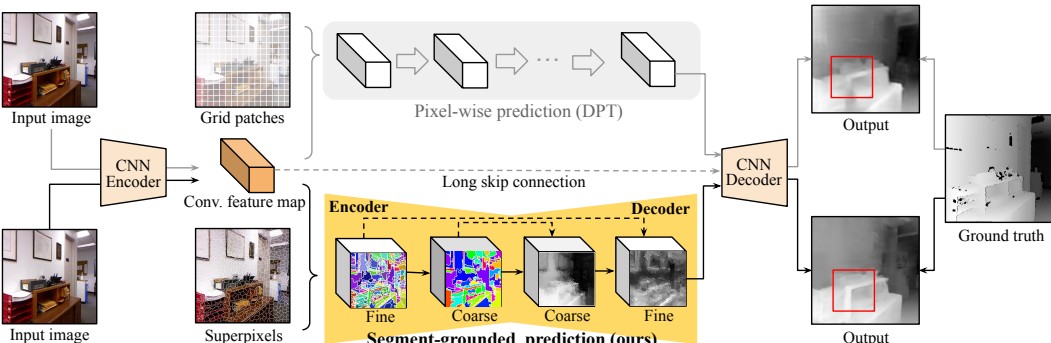

Figure 1: **Segment hierarchy for estimating depth (SHED).** Conventional methods such as DPT (Ranftl et al., 2021) perform pixel-wise prediction without considering structure, often resulting in blurry object shapes. SHED addresses this by leveraging a hierarchy of segment tokens to guide prediction. Unlike DPT, which uses fixed grid tokens across all layers, we adapt its ViT (Dosovitskiy, 2020) blocks into two stages: the encoder pools superpixel tokens into coarser segment tokens, and the decoder progressively refines predictions from coarse to fine segments, producing depth maps with structural coherence.

SHED uses a hierarchical segmentation process to define structural conditions. The encoder, which builds on the CAST (Ke et al., 2024b), a ViT-based model for hierarchical segmentation in recognition tasks, starts by representing the image as superpixels instead of standard patches. It then iteratively merges these superpixel tokens based on feature similarity, creating a multi-level hierarchy of segment tokens. To produce a structured depth, the decoder inverts this hierarchy, leveraging both the segment maps and their features. It unpools finer segments from coarser ones using soft assignments computed in the encoder, and adds them with tokens from the corresponding encoder layer. Each segment token is projected into a spatial map by distributing its features over the associated region, producing sharp boundaries across different objects and smooth transitions within the same object. The features from multiple segment levels are fused with pixel-level features from a convolutional encoder to produce outputs that preserve global layout while capturing fine detail.

We highlight the main differences between SHED and CAST (Ke et al., 2024b). First, while CAST is encoder-only, SHED extends it to an encoder-decoder for dense prediction. Second, CAST treats segmentations solely as outputs, whereas SHED also uses segment-associated features as decoder inputs to produce dense representations. Third, CAST relies on image-level supervision and produces segmentations guided by visual cues, while SHED is trained with dense supervision (e.g., depth), resulting in segmentations guided by geometric cues. Finally, CAST links reorganization to recognition in the "3Rs" (Malik et al., 2016), whereas SHED links reorganization to reconstruction.

By looping hierarchical segmentation into dense prediction, SHED offers three key advantages. **1)** Segmentation enhances depth estimation by enforcing object-level structure, yielding sharper boundaries and coherence within segments. It also achieves robust generalization in cross-domain transfer settings. **2)** Depth supervision leads to structured representations that better capture scene layout. As a result, SHED retrieves layout-similar images more accurately, increasing top-1 recall by 34% (45.2→60.5). **3)** Accurate depth maps from SHED improve 3D reconstruction, producing smooth surfaces aligned with the ground truth. Its hierarchy also enables unsupervised 3D part discovery, which DPT cannot achieve as it predicts depth holistically without structural understanding.

## 2 RELATED WORK

**Monocular depth estimation** is a representative dense prediction task, that infers per-pixel depth from a single image. It is widely used in 3D reconstruction (Song et al., 2017), autonomous driving (Geiger et al., 2012), and robotic perception (Tateno et al., 2017). Early approaches relied on hand-engineered features (Torralba & Oliva, 2002; Saxena et al., 2008), while deep learning methods later became dominant (Eigen et al., 2014; Laina et al., 2016; Godard et al., 2017; Zhou et al., 2017; Hu et al., 2019; Godard et al., 2019; Lee et al., 2019; Ranftl et al., 2020). Recent ViT (Dosovitskiy, 2020)-based models such as DPT (Ranftl et al., 2021) have shown strong performance, leveraging

foundation models pretrained on diverse data (Bhat et al., 2023; Yang et al., 2024a; Ke et al., 2024a). However, these models still struggle with structural consistency in complex scenes.

**Structural cues in depth estimation** have been extensively explored to enhance geometric coherence. Existing approaches can be broadly categorized into four types: **1)** Representation approaches modify how depth is encoded, such as by discretizing depth values (Fu et al., 2018; Bhat et al., 2021; Li et al., 2024) or modeling spatial dependencies (Liu et al., 2015; Cheng et al., 2018; Yuan et al., 2022). **2)** Regularization imposes geometric constraints through loss functions that promote smooth surfaces (Godard et al., 2017; Zhan et al., 2018; Bian et al., 2019), consistent normals (Yang et al., 2018), or planar regions (Yin et al., 2019; Watson et al., 2019). **3)** Multi-task learning jointly estimates depth with auxiliary signals, such as scene geometry (Eigen & Fergus, 2015; Yin & Shi, 2018) or semantics (Mousavian et al., 2016; Kendall et al., 2018; Chen et al., 2019; Guizilini et al., 2020; Zhu et al., 2020). **4)** Post-processing refines predictions using off-the-shelf techniques (Krähenbühl & Koltun, 2011; Chen et al., 2016).

Several multi-task approaches have explored segmentation as an auxiliary signal to improve depth estimation. Early works used segmentation as an additional supervision signal (Mousavian et al., 2016; Kendall et al., 2018), while more recent ones leveraged segment regions or boundaries to guide depth discontinuities (Chen et al., 2019; Guizilini et al., 2020; Zhu et al., 2020). SHED follows this principle but integrates segmentation and depth estimation into a unified process, enabling them to benefit from each other during training. Moreover, it discovers hierarchical segmentation in an unsupervised manner, eliminating the need for costly human annotations.

Although structural cues offer clear benefits, most existing methods do not scale well to modern architectures. Representation-based approaches often require architectural changes that are incompatible with transformers, while regularization and multi-task methods rely on additional annotations, limiting scalability. In contrast, SHED integrates seamlessly into ViT-based models such as DPT and learns structural segmentation solely from depth supervision. By design, it inherently produces sharp, segment-aligned boundaries, reducing the need for post-processing.

**Perceptual grouping** is a key mechanism in human vision that organizes low-level elements into coherent global structures (Wertheimer, 1938; Marr, 2010). This principle has inspired a broad range of computer vision research, including perception (Locatello et al., 2020; Mo et al., 2021; Kang et al., 2022; Deng et al., 2023; Ranasinghe et al., 2023), segmentation (Arbeláez et al., 2012; Hwang et al., 2019; Ke et al., 2022; Xu et al., 2022), and generation (Hong et al., 2018; Mo et al., 2018; He et al., 2022). In particular, CAST (Ke et al., 2024b) recently applied it to ViTs for concurrent segmentation and recognition. However, most of these methods, including CAST, consider only a *forward hierarchy*, constructing representations and segmentations in a bottom-up manner. In contrast, we adopt the complementary concept of a *reverse hierarchy* (Hochstein & Ahissar, 2002), where global structures guide and refine local parts through top-down feedback. We leverage this principle to design an encoder-decoder that accounts for both hierarchies.

While some prior works (Anderson et al., 2018; Shi et al., 2023; Eftekhar et al., 2023) have explored reverse hierarchies for recognition, they do not address dense prediction. Other studies (Eslami et al., 2016; Sajjadi et al., 2022; Seitzer et al., 2022) apply similar ideas to encoder-decoder architectures, but focus on object-centric representations, lacking the ability to model segment hierarchies and often producing blurry outputs. To the best of our knowledge, this is the first work to leverage bidirectional segment hierarchies to enhance dense prediction within a modern ViT framework.

## 3 SHED: SEGMENT HIERARCHY FOR ESTIMATING DEPTH

We propose SHED, which integrates a bidirectional segment hierarchy into the ViT blocks of DPT (Ranftl et al., 2021). Unlike DPT, which uses fixed-size patch tokens across all layers, our model constructs a hierarchy of segment tokens: the encoder builds a forward hierarchy by grouping features from fine to coarse, while the decoder applies a reverse hierarchy to refine predictions from coarse to fine, guided by the learned segment tokens. This design, illustrated in Figure 2, enables the model to progressively reorganize and reconstruct structured scene information.

### 3.1 ENCODER: GROUPING SEGMENTS VIA FORWARD HIERARCHY

Our encoder builds on CAST (Ke et al., 2024b), which **1)** replaces square patch tokens with superpixel tokens, and **2)** progressively clusters them into coarser segment tokens by token similarity. This process produces a fine-to-coarse hierarchy of segment tokens. CAST was originally developed

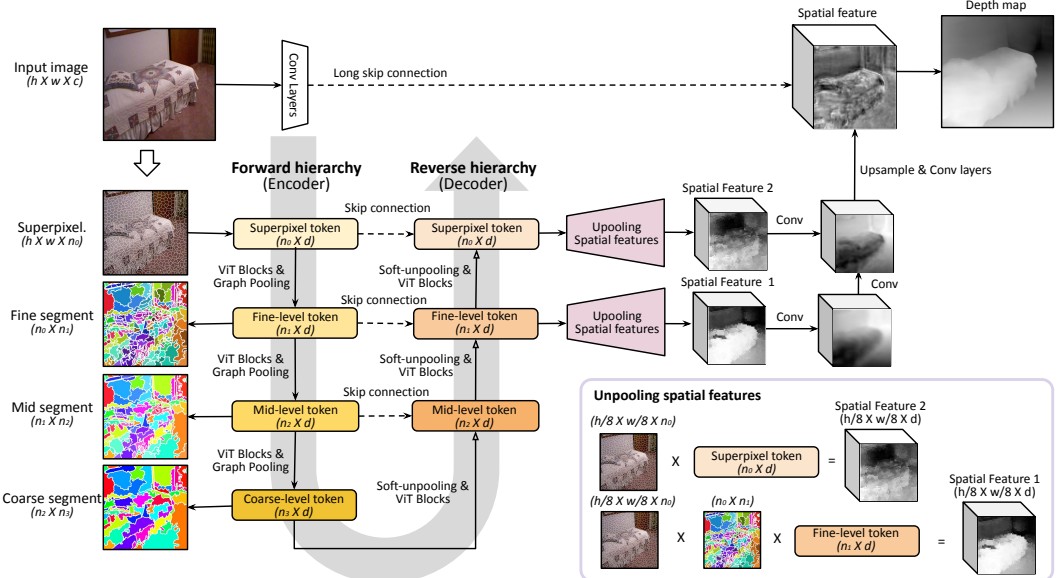

Figure 2: **SHED integrates a forward and reverse segment hierarchy into the ViT blocks.** Following the overall architecture of DPT which uses a standard decoder design choice of depth foundation models including convolutional layers for monocular depth estimation, we adapt the ViT into two stages. **1)** The encoder converts the input image into superpixel tokens and applies graph pooling to form coarser segments, following the hierarchical clustering strategy of CAST (Ke et al., 2024b). **2)** The decoder reverses this hierarchy by unpooling segment tokens from coarse to fine and fusing them with encoder features at corresponding levels via skip connections. The tokens are projected into 2D maps according to their regions. These multi-level maps are fused with pixel-level features from early convolutional layers to recover fine details and produce the final depth map.

as an encoder-only model for image-level recognition. We extend it into an encoder-decoder, where the segment hierarchy not only guides dense prediction but is also refined through dense supervision.

**Tokenization.** Given an input image $X \in \mathbb{R}^{h \times w \times c}$, the encoder produces hierarchical segmentations $S_0, S_1, \ldots$ and corresponding embeddings $Z_0, Z_1, \ldots$, ordered from fine to coarse. This process begins by dividing the image into $n_0$ superpixels, which yields a one-hot assignment matrix $S_0 \in \mathbb{R}^{(h \cdot w) \times n_0}$ that maps each pixel to a superpixel.

We extract a convolutional feature map $F_{\text{conv}} \in \mathbb{R}^{(h_0 \cdot w_0) \times d}$ with spatial stride 8 ($h_0 = h/8$, $w_0 = w/8$), add fixed sinusoidal positional embeddings, and average-pool features within each superpixel to obtain initial embeddings $Z_0 \in \mathbb{R}^{n_0 \times d}$. To enable global context modeling, we append a class token to form $\bar{Z}_0 \in \mathbb{R}^{(n_0+1) \times d}$, which is passed to the first ViT block.

**Hierarchical clustering.** We construct coarser segment tokens by alternating ViT blocks with graph pooling (Ke et al., 2024b). At each level $l$, given $Z_{l-1}$ and $S_{l-1}$ from the previous layer, we append a class token to form $\bar{Z}_{l-1}$, apply ViT blocks, and obtain updated features, excluding the class token.

To form coarser tokens $Z_l \in \mathbb{R}^{n_l \times d}$, we compute a soft assignment matrix $P_l \in \mathbb{R}^{n_{l-1} \times n_l}$ based on cosine similarity between fine- and coarse-level tokens:

$$P_l(i \rightarrow j) \propto \text{sim}(Z_{l-1}[i], Z_l[j]), \quad \text{for } i \in [n_{l-1}], \, j \in [n_l],$$

where $[n] := \{0, \ldots, n-1\}$. The coarse tokens $Z_l$ are initialized via farthest point sampling (Qi et al., 2017) from $Z_{l-1}$, and refined by aggregating fine-level features weighted by $P_l$, followed by an MLP and a residual connection:

$$Z_l \leftarrow Z_l + \text{MLP}(P_l^\top Z_{l-1} \oslash P_l^\top \mathbf{1}),$$

where $\oslash$ denotes element-wise division for normalization.

To propagate segmentation labels through the hierarchy, we compute coarser segmentations by composing the assignment matrices:

$$S_l = S_{l-1} \bar{P}_l, \quad l = 1, 2, \ldots, l_{\max},$$

where $\bar{P}_l$ is a hard assignment matrix obtained by taking the argmax over each row of $P_l$.

## 3.2 DECODER: PREDICTING OUTPUTS VIA REVERSE HIERARCHY

The decoder reconstructs spatial feature maps by reversing the encoder's segment hierarchy, progressively unpooling segment tokens $Z_{l_{\max}}, \ldots, Z_0$. This involves two steps: **1)** computing decoder features $Z_l'$ by unpooling from $Z_{l+1}'$ and fusing them with encoder features $Z_l$ via skip connections; and **2)** projecting $Z_l'$ to the image space to obtain a spatial feature map $F_l$ of size $(h_l, w_l)$.

**Unpooling segment tokens.** We reverse the encoder's clustering in a coarse-to-fine manner. At each level $l = l_{\max} - 1, \ldots, 0$, we compute

$$Z_l' \leftarrow P_{l+1}^\top Z_{l+1}',$$

which distributes coarse features to finer segments. We then add the unpooled features with the corresponding encoder output:

$$Z_l' \leftarrow \mathrm{MLP}(Z_l' + Z_l),$$

followed by ViT blocks with class tokens.

**Unpooling spatial features.** We convert the segment tokens $Z_l'$ into spatial feature maps by composing the soft assignment matrices:

$$P_{0 \to l} = P_1 \cdots P_l \quad \in \mathbb{R}^{n_0 \times n_l},$$

and applying them to the initial superpixel-to-pixel map $S_0$ to obtain soft segmentations $S_{0 \to l} = S_0 P_{0 \to l}$. The spatial feature map is then reconstructed as

$$F_l = S_{0 \to l} Z_l', \quad F_l \in \mathbb{R}^{(h_l \cdot w_l) \times d}.$$

The set of spatial maps $\{F_l\}_{l=1}^{l_{\max}}$ is fused using convolutional layers, combined with $F_{\mathrm{conv}}$, and further refined through final convolution and upsampling to produce the final dense prediction.

DPT reduces the spatial resolution of feature maps $F_l$ at each level by a factor of $2^l$, with $h_l = h_0/2^l$, $w_l = w_0/2^l$, producing coarse maps in early ViT layers that are progressively refined. This forms a *spatial hierarchy* similar to U-Net (Ronneberger et al., 2015), improving global coherence and reducing computation. However, it relies on local aggregation, which lacks fine-grained structure, and reduces computation only in the final decoder. In contrast, our *segment hierarchy* groups segment regions, providing a stronger inductive bias that promotes structural consistency and reducing computation in the ViT blocks. As a result, applying spatial downsampling in SHED was not beneficial: it yielded minimal efficiency gains in the decoder while degrading boundary quality by projecting coarse segments onto low-resolution maps. Therefore, we omit spatial reduction in SHED and simply set $h_l = h_0$, $w_l = w_0$.

## 4 EXPERIMENTS

We demonstrate the benefits of SHED by integrating segmentation into the loop for dense prediction: **1)** Segment-consistent depth estimation that preserves occlusion boundaries and intra-segment coherence, leading to improved accuracy and efficiency; **2)** Structure-aware representation learning through dense supervision, enabling layout-aware features and segmentations; **3)** 3D scene reconstruction from predicted depth maps, yielding globally coherent and part-aware structures.

### 4.1 SETUP

We implement SHED on top of DPT (Ranftl et al., 2021), adopting its overall training setup. Specifically, we use the DPT-Hybrid variant, which combines ResNet-50 (He et al., 2016) and ViT-Small (Dosovitskiy, 2020), and refer to it simply as DPT throughout the paper. For in-domain evaluation, we primarily train and evaluate on NYUv2 (Nathan Silberman & Fergus, 2012), a standard benchmark for indoor depth estimation. For cross-domain transfer, we train SHED on the synthetic HyperSim dataset (Roberts et al., 2021) and assess its zero-shot generalization on the real-world NYUv2 dataset. We further compare our approach with stronger prior-based models, including the DPT-style Depth Anything v2 (Yang et al., 2024b) and Marigold (Ke et al., 2025), both fine-tuned on HyperSim. Depth Anything v2 uses the DPT decoder with DINOv2 (Oquab et al., 2023) encoder.

**Tokenization.** Input images of size $640 \times 480$ are randomly cropped to $384 \times 384$ during preprocessing. We generate 576 superpixels using the SEEDS algorithm (Van den Bergh et al., 2012), matching

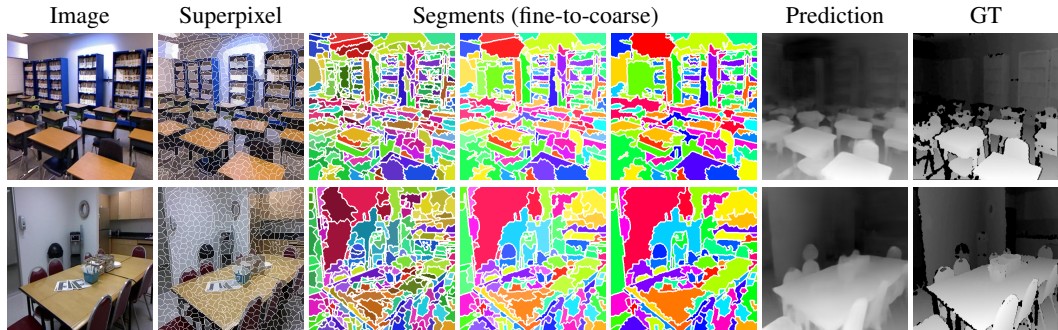

Figure 3: **SHED produces consistent structures in predicted depth map with spatio-layout.** We visualize the fine-to-coarse segments and corresponding depth maps from SHED, along with ground truth (GT) depth for comparison. Examples are from the NYUv2 test set. SHED captures fine structures through its segments, such as desks in a classroom, which allow the depth map to clearly separate them from the background (row 1). It also decomposes large objects, such as a table, into multiple parts, leading to smooth depth variations toward the back (row 2).

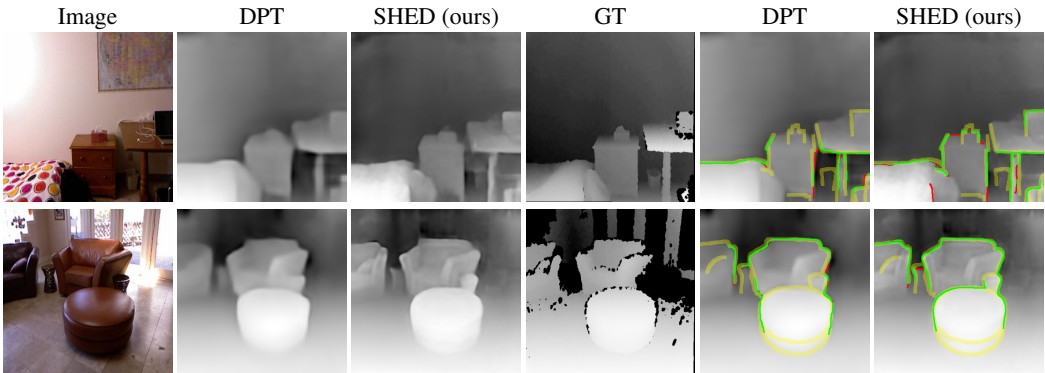

Figure 4: **SHED generates sharper object contours, clearer occlusion boundaries, and more coherent values within segments.** We compare depth maps (cols 2-4) and occlusion boundaries (cols 5, 6) from DPT, SHED on the NYUv2-OC++ dataset. Boundaries are extracted using a Canny edge detector and evaluated against GT, with GT edges shown in yellow, true positive in green and false positive in red. SHED more accurately captures object edges and produces smoother depth within segments. Its predicted boundaries also align more closely with the ground truth.

the 24×24 token grid of DPT, which corresponds to 16×16 patches. Features are extracted from intermediate ResNet-50 blocks at 1/4 and 1/8 of the input resolution; the latter initializes segment token embeddings, while both are passed to the final decoder via skip connections. This entire preprocessing and tokenization pipeline is applied consistently in all experiments.

**Architecture.** We modify the ViT encoder-decoder in DPT by inserting graph pooling and un-pooling layers. The encoder consists of three stages, each with two ViT blocks followed by graph pooling, progressively reducing the number of segment tokens to 256, 128, and 64. The decoder mirrors this with unpooling and receives skip connections from the corresponding encoder stages.

**Training.** We train both SHED and DPT on NYUv2 for a fair comparison, using a batch size of 16 for 50 epochs with the Adam optimizer (Kingma & Ba, 2014) and a learning rate of 5e-5. With pretrained ResNet and ViT backbones, we follow DPT's default training recipe, including the scale-invariant logarithmic loss computed against ground-truth depth. At inference time, predicted depth maps at 384×384 resolution are bilinearly upsampled to 640×480 to match the ground-truth size.

## 4.2 SEGMENT-CONSISTENT DEPTH ESTIMATION

SHED generates structured depth maps by leveraging a learned segment hierarchy. We begin by visualizing the hierarchy and predicted depth to illustrate their structural alignment. Next, we evaluate quality in terms of boundary accuracy and intra-segment coherence. Finally, we show that hierarchical decoding improves efficiency without compromising pixel-wise accuracy.

Table 1: **SHED improves boundary accuracy and object-wise depth accuracy and error.** We evaluate the structural quality of depth maps using two metrics: **1)** Occlusion boundary error, evaluated on the NYUv2-OC++ dataset. Occlusion boundaries are extracted using a Canny edge detector, and the Chamfer distance is computed in both directions: from prediction to ground truth and vice versa. **2)** Intra-segment coherence measures how well the predicted depth values within each object align with the ground-truth. We compute this with object-level annotations.

| Method | Boundary Error ↓ | | Object-wise Depth Accuracy ↑ | Object-wise Depth Error ↓ | | |
|---|---|---|---|---|---|---|
| | $\epsilon_a$ | $\epsilon_c$ | $\delta > 1.25$ | AbsRel | RMSE | log 10 |
| DPT | 6.395 | 1.438 | 0.802 | 0.144 | 0.500 | 0.061 |
| SHED (ours) | **5.713** | **0.608** | **0.814** | **0.142** | **0.496** | **0.060** |

Table 2: **SHED improves both in-domain and cross-domain (synthetic → real) depth estimation.** We evaluate standard depth accuracy and error metrics on the NYUv2 test set. SHED delivers competitive per-pixel depth estimation performance comparable to DPT when trained in-domain. In cross-domain zero-shot evaluation, it shows competitive generalization compared to Depth Anything v2 and Marigold.

| Method | Pre-training | Training | Depth Accuracy | | | Depth Error | | |
|---|---|---|---|---|---|---|---|---|
| | | | $\delta > 1.25 \uparrow$ | $\delta > 1.25^2 \uparrow$ | $\delta > 1.25^3 \uparrow$ | AbsRel ↓ | RMSE ↓ | log10 ↓ |
| DPT | IN-1K | NYUv2 | 0.839 | 0.971 | **0.992** | 0.132 | 0.457 | 0.055 |
| SHED (ours) | IN-1K | NYUv2 | **0.846** | **0.972** | 0.992 | **0.130** | **0.451** | **0.054** |
| Marigold | Laion-5b | HyperSim | 0.375 | 0.659 | 0.833 | 0.542 | 1.243 | 0.171 |
| Depth Anything v2 (small) | LVM-142M | HyperSim | 0.592 | **0.902** | 0.960 | 0.749 | 0.808 | 0.110 |
| Depth Anything v2 (base) | LVM-142M | HyperSim | 0.346 | 0.889 | **0.985** | **0.341** | 0.898 | 0.123 |
| SHED (ours) | IN-1K | HyperSim | **0.632** | 0.892 | 0.960 | 0.583 | **0.740** | **0.102** |

Figure 3 shows that the segment hierarchy in SHED yields depth maps with coherent object geometry. The learned segments capture contours of objects, such as desks in a classroom, allowing the depth to clearly separate them from the floor. They also decompose larger structures, like tables, into parts, enabling smooth depth transitions from front to back. This suggests that structure guides depth prediction toward more accurate and interpretable results.

**Boundary accuracy.** We assess the structural quality of SHED by comparing its boundary predictions to those of DPT for in-domain evalution. Figure 4 shows predicted depth maps and their occlusion boundaries, extracted using a Canny edge detector (Canny, 1986), on samples from the NYUv2-OC++ dataset (Ramamonjisoa et al., 2020). For quantitative evaluation, we follow the standard protocol (Koch et al., 2018) and compute the average Chamfer distance (Fan et al., 2017) in two directions: from prediction to ground truth, and vice versa. SHED produces sharper contours and outperforms DPT on both metrics, with particularly large gains in recall, likely due to its fine-grained segmentation. However, oversegmentation may introduce spurious edges that reduce precision, highlighting the importance of accurate segmentation.

**Intra-segment coherence.** Beyond boundary, we evaluate how coherently depth values vary within each segment. We use a metric called object-wise depth accuracy and error, which measures the pixel-wise depth accuracy and error between the predicted and ground-truth depth depth maps within each segment, treating the latter as structural references. As shown in Figure 4, SHED produces smoother depth variations within segments. This is reflected quantitatively in Table 1.

**Per-pixel metrics.** We compare SHED with DPT for the evaluation of the in-domain and Depth Anything v2 and Marigold for the evaluation of cross-domain transfer using standard depth metrics per pixel, as shown in Table 2. In in-domain evaluation, SHED shows competitive per-pixel performance compared to DPT. in the cross-domain evaluation, SHED demonstrates superior transfer capabilities by outperforming both Depth Anything v2 (Yang et al., 2024b) which leverages the strong DINOv2 (Oquab et al., 2023) encoder pre-trained on over one million images and Marigold (Ke et al., 2025) which is pre-trained on over five billion images in the majority of metrics, highlighting SHED's effectiveness despite using comparatively less pre-training data.

Table 3 includes additional results under mixed-training settings to assess robustness across different domains. To further strengthen the experimental evaluation, we conduct new zero-shot cross-domain evaluation in which SHED is trained exclusively on multiple datasets, including HyperSim (Roberts et al., 2021), vKiTTI2 and MegaDepth (Li & Snavely, 2018) and evaluated without any fine-tuning on multiple real-world benchmarks including KiTTI, NYUv2, SUN-RGBD.

Table 3: **SHED improves mixed data setting.** We evaluate standard depth accuracy and error metrics on more diverse dataset. SHED delivers competitive per-pixel depth estimation performance comparable to DPT when trained mixed training setting in cross-domain zero-shot evaluation.

| Method | KITTI | | NYUv2 | | SUN-RGBD | |
|---|---|---|---|---|---|---|
| | AbsRel $\downarrow$ | $\delta > 1.25 \uparrow$ | AbsRel $\downarrow$ | $\delta > 1.25 \uparrow$ | AbsRel $\downarrow$ | $\delta > 1.25 \uparrow$ |
| DPT | 0.286 | 0.475 | 0.247 | 0.559 | 8.58 | 0.169 |
| SHED (ours) | **0.272** | **0.500** | **0.244** | **0.571** | **7.16** | **0.190** |

Table 4: **Ablation study.** We evaluate depth accuracy and error metrics on the NYUv2 test set.

| Method | Depth Accuracy | | | Depth Error | | |
|---|---|---|---|---|---|---|
| | $\delta > 1.25 \uparrow$ | $\delta > 1.25^2 \uparrow$ | $\delta > 1.25^3 \uparrow$ | AbsRel $\downarrow$ | RMSE $\downarrow$ | log10 $\downarrow$ |
| DPT | 0.839 | 0.971 | 0.992 | 0.132 | 0.457 | 0.055 |
| SHED with forward hierarchy only | 0.755 | 0.951 | 0.989 | 0.163 | 0.552 | 0.069 |
| SHED (ours) | **0.846** | **0.972** | **0.992** | **0.130** | **0.451** | **0.054** |

Table 5: **SHED scales outperforms segmentation-guided depth estimation.** We evaluate standard depth accuracy and error metrics on the NYUv2 test set. These include approaches that integrate over-segmentation to impose object-level consistency (Simsar et al., 2022) and classical multi-task models (Mousavian et al., 2016) that jointly predict depth and semantics.

| Method | Depth Accuracy | | | Depth Error | | |
|---|---|---|---|---|---|---|
| | $\delta > 1.25 \uparrow$ | $\delta > 1.25^2 \uparrow$ | $\delta > 1.25^3 \uparrow$ | AbsRel $\downarrow$ | RMSE $\downarrow$ | log10 $\downarrow$ |
| Simsar et al. | 0.847 | 0.971 | 0.993 | 0.116 | 0.448 | - |
| Mousavian et al. | 0.568 | 0.856 | 0.956 | 0.200 | 0.816 | 0.061 |
| SHED (ours) | **0.855** | **0.974** | **0.993** | **0.123** | **0.433** | **0.052** |

**Computational efficiency.** For a fair and hardware-independent comparison of structural complexity, we report both the number of parameters and the computational cost in FLOPs. DPT-Hybrid contains 41.88 million parameters and requires 135.0 GFLOPs. In contrast, SHED uses 56.58 million parameters but reduces the computation to 103.2 GFLOPs, which is approximately a 24% decrease in FLOPs. This substantial reduction demonstrates that SHED is structurally more efficient and achieves lower theoretical latency despite having a slightly larger parameter count.

**Segmentation-guided depth estimation.** In addition to ViT-based depth foundation models, Table 5 shows comparison results of SHED against other methodologies that utilize structural cues, specifically those employing segmentation to enhance depth estimation. There is an approach to integrate over-segmentation into the depth network to enforce object-level consistency (Simsar et al., 2022). Classical multi-task learning approach (Mousavian et al., 2016) also predicts depth and semantics jointly, using semantic boundaries to guide pixel-wise depth refinement. While effective in constrained settings, these approaches rely on task-specific supervision or post-processing pipelines, making them difficult to scale naturally to modern transformer-based architectures.

**Ablation.** We conducted an ablation study to isolate the contributions of the hierarchical clustering in the encoder and the hierarchy reversing in the decoder. To verify the necessity of our proposed decoder, we experimented with a variant of SHED by removing the progressive unpooling process in the decoder. Table 4 shows removing the reverse hierarchy leads to a sharp performance degradation, falling significantly behind the DPT baseline. This demonstrates that the coarse-to-fine unpooling mechanism in the decoder is essential to recover fine-grained spatial details from the grouped representations.

### 4.3 STRUCTURE-AWARE REPRESENTATION LEARNING

Our architecture not only improves depth prediction but also facilitates structure-aware representation learning. First, SHED learns features that reflect scene layout, enabling more accurate layout-aware image retrieval than DPT (Ranftl et al., 2021). Second, its segment hierarchy captures geometric cues informed by depth supervision, whereas CAST (Ke et al., 2024b) relies on visual cues.

**Layout-aware image retrieval.** We assess the structural understanding of learned representations by performing layout-aware image retrieval on the NYUv2 dataset, using 120K video frames collected from 206 scenes. These frames serve as queries, and we define two retrieval settings. In scene

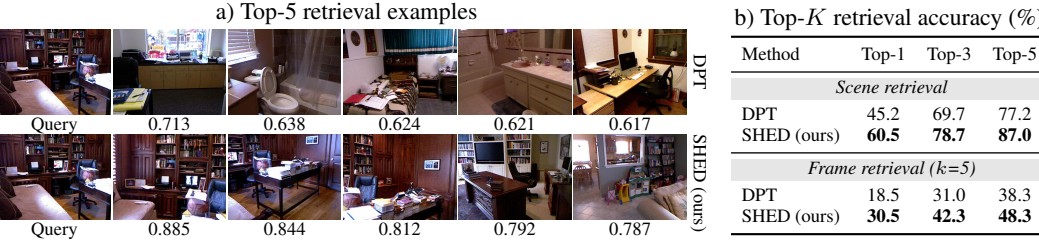

a) Top-5 retrieval examples

b) Top-K retrieval accuracy (%)

| Method | Top-1 | Top-3 | Top-5 |
|---|---|---|---|
| *Scene retrieval* | | | |
| DPT | 45.2 | 69.7 | 77.2 |
| SHED (ours) | **60.5** | **78.7** | **87.0** |
| *Frame retrieval (k=5)* | | | |
| DPT | 18.5 | 31.0 | 38.3 |
| SHED (ours) | **30.5** | **42.3** | **48.3** |

Figure 5: **SHED learns layout-aware representations through depth supervision.** We evaluate image retrieval on NYUv2 based on cosine similarity between class tokens from the final ViT block. **a)** Top-5 results (ranked left to right), with similarity scores shown below. SHED retrieves images with similar layouts, such as a central desk and a rear bookshelf, while DPT retrieves unrelated scenes. **b)** Top-$K$ accuracy at the scene and frame level ($k = 5$), where the targets are different views from the same scene or nearby frames. SHED significantly outperforms DPT in all settings, indicating that our depth-guided segmentation effectively encodes spatial layout.

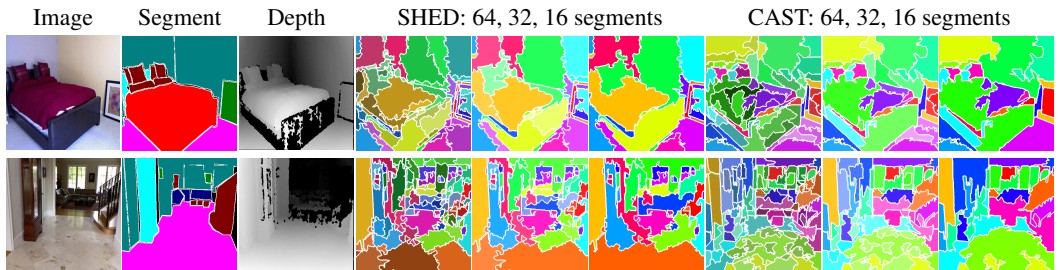

Image Segment Depth SHED: 64, 32, 16 segments CAST: 64, 32, 16 segments

Figure 6: **SHED learns depth-aware segment hierarchies, while CAST relies on visual cues.** We compare segmentations from SHED and CAST (Ke et al., 2024b) at the same hierarchy levels: 64, 32, and 16 segments. SHED captures meaningful part structures, such as separating the blanket and pillow from the bed (row 1). It also decomposes large structures like the floor based on depth, grouping nearby regions into a single large segment while dividing distant areas into smaller ones (row 2). In contrast, CAST relies on appearance cues and fails to capture geometric structure. For instance, it groups white floor regions by color but divides them arbitrarily, ignoring depth. These results highlight the value of depth supervision in learning 3D-aware segmentations.

retrieval, all frames from the same sequence are valid targets. For finer-grained evaluation, we also consider frame-$k$ retrieval, where only frames within $k$ time steps of the query are included. Given a query image, we rank other images by the cosine similarity of their class tokens from the final ViT decoder block. Figure 5 presents both qualitative and quantitative results. The left side shows that SHED retrieves images with similar spatial layouts, such as a central desk and a rear bookshelf, while DPT returns unrelated scenes. The right side shows that SHED significantly outperforms DPT in both scene- and frame-level metrics, improving Top-1 recall in scene retrieval from 45.2 to 60.5.

**Depth-aware image segmentation.** We analyze the segment hierarchy learned by SHED by comparing it to CAST, an encoder trained for image recognition using segment-based representations. We use CAST-B, trained on ImageNet (Deng et al., 2009) with the MoCo-v3 objective (Chen et al., 2021), a self-supervised learning by instance discrimination (Wu et al., 2018) that clusters visually similar images. Following CAST's setup, we use 224×224 images and extract 196 superpixels, clustered into 64, 32, and 16 segments. For fairness, we produce the same number of segments by adapting the graph pooling layers of SHED, keeping the original input resolution and superpixels.

Figure 6 shows qualitative results. SHED learns hierarchical structures that align with scene geometry: it separates objects like blankets and decomposes large structures such as floors into segments that reflect their spatial extent. In contrast, CAST groups regions based on appearance. For example, it clusters white floor areas by color but fails to account for geometric cues. We attribute this difference to the training objective: CAST learns segments through image-level recognition, while SHED is guided by dense prediction. Although our focus here is depth, the ability to learn segment hierarchies grounded in 3D structure opens possibilities for other dense prediction tasks as well.

We additionally evaluate the quality of the learned segment hierarchy using standard segmentation metrics. Following CAST, we compute boundary F-score and region IoU between the predicted

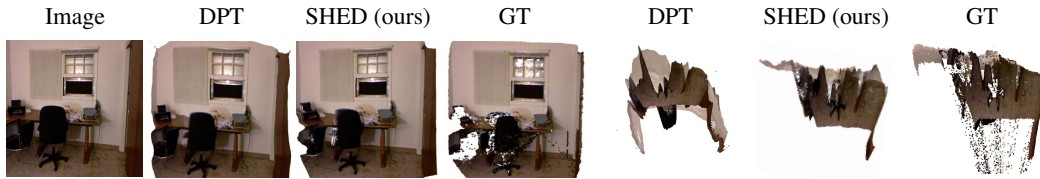

Figure 7: **SHED produces structured 3D reconstructions.** We visualize 3D point clouds reconstructed from single-view depth maps, following the semantic scene completion protocol (Song et al., 2017), using predictions from DPT, SHED, and the ground truth on NYUv2 examples. Frontal views (cols 2-4) show that DPT fails to preserve planar structures, producing curved wall boundaries, whereas SHED more accurately recovers straight lines. This difference is more apparent in the bird's-eye views (cols 5-7): DPT yields warped surfaces, while SHED produces flatter layouts.

Table 6: **SHED improves 3D alignment.** We compute the Chamfer distance between point clouds from the predicted and ground-truth depths. SHED achieves lower errors than DPT.

| Method | Precision / Recall ↓ |
|---|---|
| DPT | 0.171 / 0.251 |
| SHED (ours) | **0.158 / 0.244** |

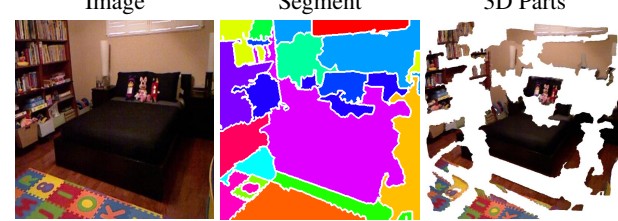

Figure 8: **SHED discovers 3D part structures.** Concurrent segmentation and depth estimation enable part-level decomposition of the reconstructed 3D point clouds.

segments and ground-truth superpixels. CAST achieves an mIoU of 43.1 and a boundary F-score of 36.5. In comparison, SHED attains an mIoU of 44.5 and a boundary F-score of 37.7, outperforming CAST on both region accuracy and boundary quality. This highlights that SHED not only leverages the hierarchy effectively for depth prediction but also learns segment boundaries that correspond well to meaningful scene geometry.

### 4.4 3D SCENE RECONSTRUCTION WITH PART STRUCTURES

We conclude by demonstrating SHED's capability for 3D scene understanding. While plausible pixel values may suffice for 2D depth estimation, accurate and structured depth is particularly critical when projected into 3D space. Accordingly, SHED enables high-quality 3D reconstruction and supports unsupervised 3D part discovery through concurrent segmentation.

To evaluate the structural quality of predicted depth maps, we project them into 3D point clouds on the NYUv2 dataset (Nathan Silberman & Fergus, 2012), following the semantic scene completion protocol (Song et al., 2017) and using NYUv2 camera intrinsics. For interpretability, all depth values are scaled by 1/1000. Figure 7 shows that SHED produces cleaner reconstructions with sharper boundaries and flatter surfaces that better align with ground truth geometry, whereas DPT yields curvier, less faithful shapes. We quantify reconstruction performance with the Chamfer distance (Fan et al., 2017) in both directions. Table 6 shows that SHED consistently achieves lower distances than DPT, confirming its advantage in structured 3D prediction. By jointly predicting segmentation and depth, SHED lifts 2D parts into 3D space, enabling part-level decomposition of scenes. Figure 8 shows an example from NYUv2, where segments corresponding to objects form coherent 3D structures in point clouds. This demonstrates SHED's potential for unsupervised 3D part reasoning, a key capability for interactive and dynamic scene understanding (Mo et al., 2019).

## 5 CONCLUSION

We shed light on the role of segmentation in depth estimation. SHED learns a segment hierarchy in the encoder and reverses it in the decoder to predict dense maps. This results in depth maps with segment-consistent structure, layout-aware representations, and coherent 3D scenes with interpretable parts. Our principle of unifying reconstruction and reorganization offers a new direction for 3D vision and robotics, particularly for tasks that require fine-grained interaction with physical components. Additional results and limitations are discussed in Section D.

**Ethics statement.** This research was conducted responsibly based on the principles outlined in the ICLR Code of Ethics. This technology can enhance the 3D environmental perception of autonomous

driving systems, thereby improving road safety, and can help robots interact more safely and efficiently with their surroundings. Before deploying this model in real-world scenarios, it must undergo rigorous and thorough validation for robustness and safety across a wide range of conditions.

**Reproducibility statement.** Appendix provides implementation details. Full release of the code upon acceptance.

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
