CONTENTS

# A  IMPLEMENTATION DETAILS

## A.1  TRAINING DETAILS

We train our model on the NYUv2 dataset (Nathan Silberman & Fergus, 2012) using the official training split. Each RGB image is first cropped to remove invalid boundaries (coordinates: 43, 45, 608, 472), then resized to $384 \times 384$ resolution. The corresponding depth maps undergo the same spatial preprocessing and are normalized by dividing raw depth values by 1000.

For data augmentation, we apply horizontal flipping with a probability of 0.5, gamma correction with $\gamma \in [0.9, 1.1]$, brightness scaling using a random factor from $[0.75, 1.25]$, and per-channel color jittering with multiplicative factors in $[0.9, 1.1]$. After augmentation, random spatial crops of size $384 \times 384$ are applied to both images and depth maps.

For tokenization, we generate superpixels using OpenCV's SEEDS (Van den Bergh et al., 2012) algorithm. Each image is segmented into 676 superpixels using a single-level hierarchy (`num_levels=1`) and a histogram bin size of 5. The algorithm is run for 50 iterations to refine superpixel boundaries.

## A.2  EVALUATION DETAILS

We evaluate on the official NYUv2 (Nathan Silberman & Fergus, 2012) test split, which contains 654 images. All evaluations use an input resolution of $384 \times 384$ pixels, with depth values clamped to the range $[10^{-3}, 10.0]$.

**Per-pixel depth metrics.** We compute standard depth estimation metrics over valid pixels where ground truth depth is available. Depth error metrics include AbsRel (mean absolute relative error), RMSE (root mean squared error), and Log10 (mean absolute logarithmic error). Accuracy is measured using threshold metrics $\delta^1$, $\delta^2$, and $\delta^3$, which denote the percentage of pixels where the predicted-to-ground-truth depth ratio is below $1.25$, $1.25^2$, and $1.25^3$, respectively. All metrics are computed with numerical safeguards, including epsilon clamping at 1e-6 to prevent division by zero and log-domain errors.

**Occlusion boundary.** We follow the evaluation protocol of the NYUv2-OC++ dataset (Ramamonjisoa et al., 2020) to assess occlusion boundary accuracy. Each predicted depth map is first min-max normalized, followed by the application of the OpenCV Canny edge detector (Canny, 1986) with low and high thresholds of 100 and 200 to produce a binary mask of predicted boundary pixels. Using the ground-truth boundary labels from NYUv2-OC++, we compute two metrics: $\varepsilon_a$ (accuracy), the average distance from each predicted edge pixel to the nearest ground-truth edge; and $\varepsilon_c$ (consistency), the average distance from each ground-truth edge pixel to the nearest predicted edge. Both are reported in squared pixels, where lower values indicate better alignment, and 0 denotes perfect correspondence.

**3D scene reconstruction.** We reconstruct 3D point clouds by back-projecting each pixel of the predicted depth maps into 3D space using the known camera intrinsics from the NYUv2 dataset, following the standard protocol for semantic scene completion (Song et al., 2017). To evaluate reconstruction quality, we compute the Chamfer distance between the predicted and ground truth point clouds. Recall quantifies how well the predicted points cover the ground truth surface, while precision measures how accurately the predicted points align with the true geometry.

Detailed evaluation protocol for **layout-aware retrieval** is provided in Section B, respectively.

## B    LAYOUT-AWARE RETRIEVAL

We provide a detailed explanation of our proposed metric, *layout-aware retrieval*, which evaluates the structural quality of learned representations by measuring how well the model retrieves frames from the same 3D scene in a video, either across the full sequence or within nearby frames.

To compute this metric, we extract the `[CLS]` token from the final layer of the vision transformer in DPT and SHED for each image, denoted as $\mathbf{z}_{\mathrm{cls}} \in \mathbb{R}^D$, and apply $l_2$ normalization. We then construct a full pairwise similarity matrix, where each entry is computed as the cosine similarity between $l_2$-normalized image embeddings: $S_{ij} = \mathbf{z}_{\mathrm{cls}}^{(i)} \cdot \mathbf{z}_{\mathrm{cls}}^{(j)}$.

Retrieval performance is evaluated in two settings: *scene-level* and *frame-level*. In the scene-level setting, the goal is to retrieve other frames from the same annotated scene, testing the model's ability to maintain structural consistency under varying viewpoints. In the frame-level setting, each frame is treated as an independent query, focusing on retrieving visually similar frames regardless of scene membership. We additionally define a frame-$k$ variant, where retrieval is restricted to the $k$ temporally adjacent frames, allowing us to assess the model's sensitivity to local layout changes. We report top-$K$ nearest neighbor retrieval accuracy on the NYUv2 (Nathan Silberman & Fergus, 2012) test set.

To further evaluate the robustness of layout-aware retrieval, we vary both the candidate set size ($k$) and the top-$K$ threshold. A smaller $k$ imposes a stricter constraint, requiring the model to identify the most similar frame from a limited pool. As shown in Figure 9, SHED consistently outperforms DPT across all settings. Performance improves for both methods as $k$ increases, with the largest gap observed at low top-$K$ values. These results suggest that SHED captures layout similarity more precisely and excels at retrieving the most relevant match.

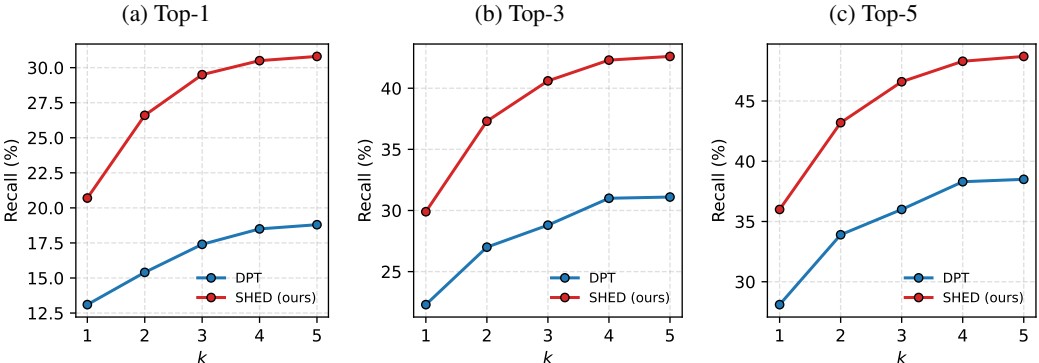

Figure 9: **Frame-$k$ recall with varying temporal range** $k$. We report Top-$K$ retrieval recall ($K \in \{1, 3, 5\}$) for DPT and SHED across temporal ranges $k \in [1, 5]$. The consistent gains highlight the robustness of our depth-supervised representation to spatial and viewpoint changes.

## C ADDITIONAL VISUALIZATIONS

### C.1 MORE VISUALIZATIONS OF DEPTH MAPS

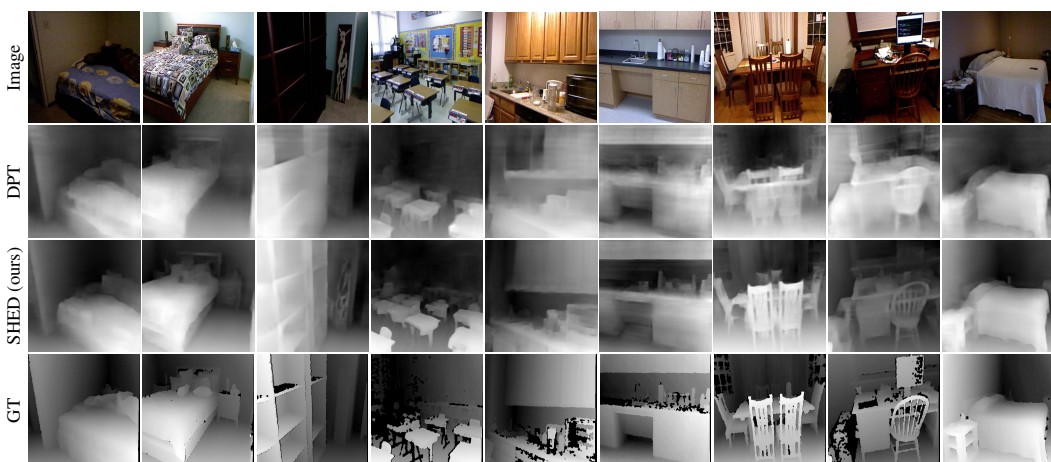

Figure 10: **Comparison of depth maps.** SHED captures more accurate shapes with sharper boundaries, whereas DPT produces blurrier results.

## C.2 MORE VISUALIZATIONS OF IMAGE RETRIEVAL

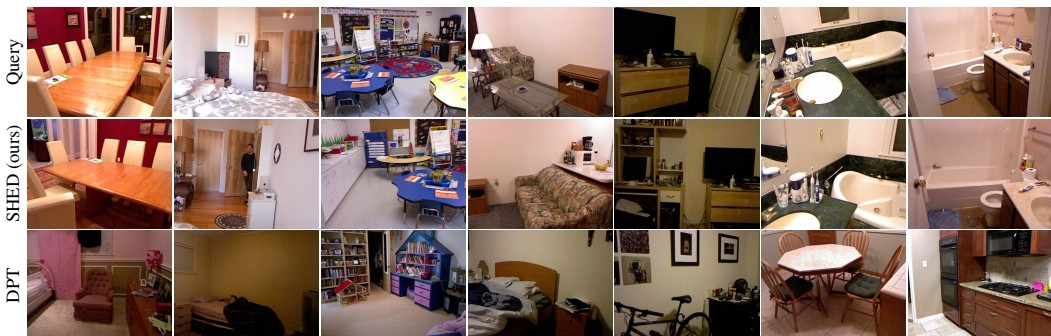

Figure 11: **Comparison of top-1 image retrieval results.** SHED retrieves samples that are more structurally similar to the query, indicating that its global embedding effectively captures scene layouts. In contrast, DPT focuses more on visual appearance, as shown in column 3, where it retrieves an image from a different scene that shares a similar color of blue.

## C.3 MORE VISUALIZATIONS OF 3D RECONSTRUCTION

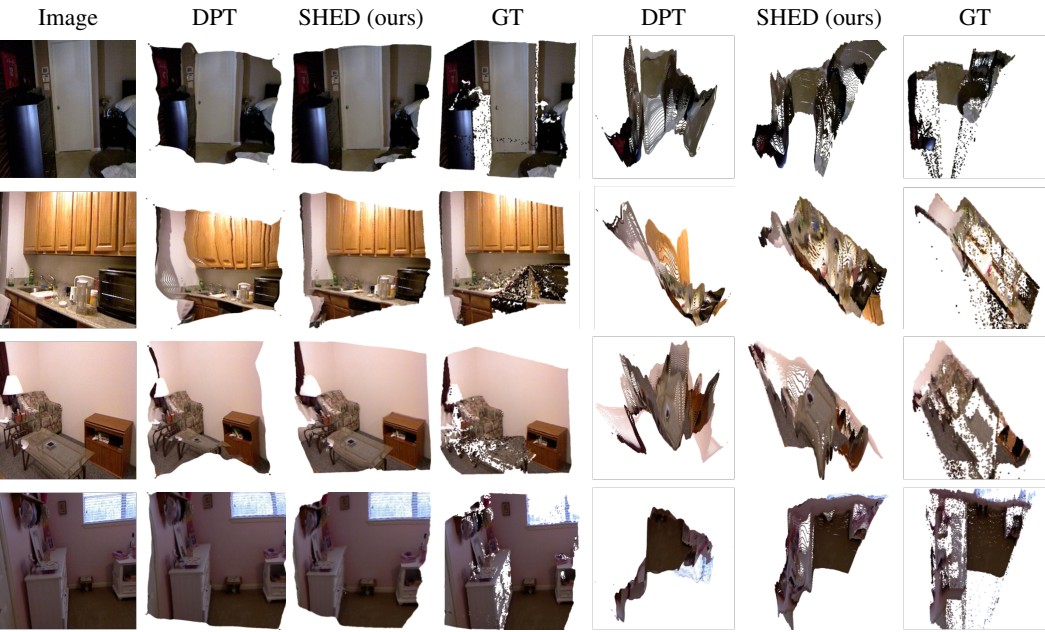

Figure 12: **Comparison of 3D reconstruction results.** Frontal views (cols 2–4) and bird's-eye views (cols 5–7). DPT yields curved wall boundaries in the frontal views, which lead to distorted 3D reconstructions visible in the bird's-eye views. In contrast, SHED produces sharp depth edges that preserve straight object contours and more faithfully represent the ground-truth 3D geometry.

## D  LIMITATIONS AND BROADER IMPACTS

### D.1  LIMITATIONS AND FUTURE WORKS

While our experiments focus on monocular depth estimation, the SHED framework is also applicable to other dense prediction tasks such as segmentation, optical flow, and image generation. We show that depth supervision induces emergent structural representations, including geometry-aware features and segmentations. Extending this approach to other objectives may uncover new forms of structure. For example, training on optical flow could yield motion-aware segmentations.

Depth estimation is fundamental to 3D scene understanding. Although we present initial results on 3D reconstruction, further investigation is needed to assess the utility of SHED in downstream tasks involving 3D reasoning and robotics. In particular, our framework enables unsupervised discovery of 3D object parts, which may serve as building blocks for modeling interactions, dynamics, and part affordances in physical environments and embodied systems.

Finally, while we compare SHED fairly against DPT under matched training conditions, state-of-the-art models such as Depth Anything benefit from large-scale pretraining and extensive engineering. Scaling up SHED with broader datasets and more compute would be a valuable next step. Notably, by jointly modeling segmentation and depth, SHED has the potential to evolve into a unified foundation model that combines the capabilities of Segment Anything and Depth Anything.

### D.2  BROADER IMPACTS

Structured understanding of the 3D world and accurate depth estimation are central challenges in AI, with direct impact on safety-critical applications such as autonomous driving, augmented reality, and robotics. In practice, failures in these systems often result not from a lack of data but from insufficient structured reasoning, making predictions vulnerable to occlusion, unusual viewpoints, and dynamic environments. Our framework promotes geometry-aware perception by producing robust and interpretable depth estimates that align with scene structure. This can improve reliability in complex real-world settings and lead to systems with more transparent failure modes.