# OpenReview forum: "SHED Light on Segmentation for Monocular Depth Estimation"
_ICLR.cc/2026/Conference — Submitted to ICLR 2026_

### Official Review · Reviewer_o4eZ · 2025-10-19

**Soundness:** 3
**Presentation:** 3
**Contribution:** 2
**Rating:** 4
**Confidence:** 4

**Summary:**

This paper ensembles the CAST into the DPT architecture with a proposed decoder (Reverse hierarchy). SHED forms superpixel tokens that are hierarchically pooled in the encoder and unpooled in the decoder. The hierarchy is learned without segmentation labels, supervised only by the final depth loss. Experiments show the advantages of this SHED. The authors provided a quite comprehensive analysis based on the resutls.

**Strengths:**

- The paper is well-motivated. Ensembling segmentation process in monocular depth estimation sounds reasonable to me. The idea of introducing CAST is also natural from my point of view.

- Layout-aware retrieval evaluation is interesting. This metric provides a reasonable way to evaluate the gain from adding CAST.

- Large gains in boundary/part structure can somehow demonstrate the effectiveness of this method.

**Weaknesses:**

- The technical contribution can be incremental. The major point is to adopt CAST for depth estimation, with merely one decoder proposed.

- Comparative breadth and fairness. Outdoor / diverse domains: Experiments focus on NYUv2 (indoor). It remains unclear if the segment-hierarchy inductive bias retains benefits on outdoor benchmarks

- It would be better to add DPT results in Table 2 (zero-shot). Is it possible to align the encoder of DAV2 when comparing? The zero-shot performance difference can be derived by the encoder.

- Runtime, memory, and scalability. The method adds CAST into DPT. There are qualitative claims about efficiency, but no measured throughput, VRAM, or FLOPs relative to DPT/DAV2

- Positioning vs. multi-task & post-processing baselines. While SHED is not for a classical multi-task learning, the most direct competitors, depth+semantics or edge-guided methods, are missed in the paper.

- When leaking the segmentation GT, it would be good to apply such a weak supervision. But the paper majorly focuses on NYU, a small dataset with segmentation labels. I was wondering if the method can be scaled up, using more training data.

**Questions:**

Please check the weakness.

---

> ### Author Response · Authors · 2025-11-21
>
> Dear reviewer o4ez,
>
> Thank you for highlighting the strengths of our work. We appreciate your recognition that the paper is well-motivated and that incorporating a segmentation process into monocular depth estimation is a reasonable. Additionally, we appreciate your positive remarks on our layout-aware retrieval evaluation, as well as comments that the large gains in boundary and part-level structure demonstrate the effectiveness of the proposed method. We address your concerns and questions in detail below.
>
> **[W1] Separation of contributions from CAST**
>
> We clarify that SHED is not a mere combination of existing modules but a novel architectural extension that transforms the encoder-only CAST framework, originally designed for recognition, into a symmetric encoder-decoder structure capable of dense prediction. Our proposed reverse hierarchy decoder uniquely unpools abstract segment tokens in progressive way to reconstruct pixel-wise geometry, a mechanism absent in prior work. Regarding performance, since we strictly controlled the backbone for a fair comparison, the observed gains are entirely attributable to our method. Furthermore, SHED demonstrates significant structural superiority, evidenced by sharper boundaries, robust zero-shot generalization outperforming Depth Anything v2, and a 34% improvement of top-1 recall in layout-aware scene retrieval.

---

> ### Author Response · Authors · 2025-11-21
>
> **[W2 & W3a] Evaluation on diverse benchmarks**
>
> we extend our experiments to a mixed-domain training setup using Virtual KITTI, HyperSim, and MegaDepth [1]. We train the SHED using a uniform sampling strategy across all datasets for 20 epochs with batch size of 64. We then evaluate this model in a fully zero-shot manner on four unseen datasets, including KITTI (outdoor), NYUv2 (indoor), SUN-RGBD (indoor).
>
> In more scalable setting, the results below demonstrate that SHED consistently outperforms the DPT baseline across domains, including both indoor and outdoor scenes. The segment-hierarchy inductive bias of SHED is not specific to the NYUv2 domain but transfers effectively to outdoor and mixed-domain settings, consistently outperforming the baseline.
>
> | Method        | KITTI AbsRel $\downarrow$ | KITTI $\delta$>1.25 $\uparrow$ | NYUv2 AbsRel $\downarrow$  | NYUv2 $\delta$>1.25 $\uparrow$ | SUN-RGBD AbsRel $\downarrow$  | SUN-RGBD $\delta$>1.25 $\uparrow$  |
> |---------------|-----------------|-----------------|-----------------|-----------------|--------------------|--------------------|
> | DPT           | 0.286               | 0.475               | 0.247               | 0.559              |           8.58        | 0.169                  |
> | SHED (ours)  | **0.272**              | **0.500**           |  **0.244**              | **0.571**               | **7.16**                  | **0.190**                  |
>
> [1] Megadepth: Learning single-view depth prediction from internet photos (CVPR 18, Li, Zhengqi)
>
>
> **[W3b] Comparison with stronger baselines**
>
> The table compares SHED with both DAV2-small and DAV2-base, where DAV2-base is a substantially stronger backbone (97.5M params, LVM-142M pre-training). Despite this difference in capacity and pre-training scale, SHED demonstrates competitive generalization performance from HyperSim to NYUv2.
>
> \begin{array}{l|llcccccc}
> \text{Method}  & \text{Params (M)} & \text{Pre-training} & \text{Training} & \delta > 1.25 \uparrow  & \delta >  1.25^2 \uparrow &  \delta >  1.25^3 \uparrow &\text{AbsRel} \downarrow &\text{RMSE} \downarrow & \log10 \downarrow  \newline
> \hline
> \text{Depth Anything 2 (small)} & 24.8 & \text{LVM-142M} & \text{HyperSim} & 0.592 & \textbf{0.902} & 0.960 & 0.749 & 0.808 &  0.110  \newline
> \text{Depth Anything 2 (base)} & 97.5 & \text{LVM-142M} & \text{HyperSim} & 0.346 & 0.889 & \textbf{0.985} & \textbf{0.341} & 0.898 &  0.123  \newline
> \text{SHED (ours)}  & 56.5 & \text{IN-1K} & \text{HyperSim} & \textbf{0.632} & 0.892 & 0.960 & 0.583 & \textbf{0.740} & \textbf{0.102}
> \newline
> \end{array}
>
> **[W4] Efficiency**
>
> We report parameters and FLOPs for a fair, hardware-independent comparison of structural complexity. As shown in the table below, SHED reduces the computational cost GFLOPs by approximately 24% compared to DPT-Hybrid, indicating superior structural efficiency and lower theoretical latency. Although our method has a higher parameter count, the lower GFLOPs indicate that SHED is structurally more efficient during inference.
>
> Importantly, we also include comparisons with Depth Anything 2 (DAV2), which serves as a strong modern baseline. While DAV2-small is lightweight in parameter count, its FLOPs (57.5G) remain comparable to DPT-Hybrid and significantly higher than what would be expected from its model size. Likewise, DAV2-base incurs the highest computational cost (201.1G FLOPs) among all methods. In contrast, SHED operates with substantially lower FLOPs than DAV2-base while achieving competitive generalization results in cross-domain setting, highlighting that SHED’s performance does not rely on capacity of the model.
>
> | Method | Params (M) | FLOPs (G) |
> | :--- | :---: | :---: |
> | Depth Anything 2 (small) | 24.8 | 57.5 |
> | Depth Anything 2 (base) | 97.5 | 201.1 |
> | DPT-Hybrid | 41.88 | 135.0 |
> | SHED (ours) | 56.58 | 103.2 |

---

> ### Author Response · Authors · 2025-11-21
>
> **[W5] Separation of contributions from joint semantic segmentation-depth  approaches**
>
> SHED differs from existing segmentation-assisted depth estimation approaches in several fundamental ways.
>
> SHED adopts a fundamentally different perspective on segmentation. In our work, segmentation is not an external auxiliary signal provided by off-the-shelf models or human annotations. Instead, it is a learned structural representation that emerges entirely from depth supervision. SHED discovers a bidirectional segment hierarchy internally without any explicit segmentation labels.
>
> In particular, the resulting segments differ qualitatively from semantic segmentation as our hierarchy is learned via depth reconstruction. As shown in Fig. 6, SHED decomposes large surfaces (e.g., a floor) based on depth variations and groups spatially consistent regions. This reveals a geometry-aware organization that standard appearance-based or semantic segmentation methods fail to capture.
>
> Our results also demonstrate that this learned hierarchy significantly improves cross-domain generalization. Compared to DPT and Depth Anything V2, SHED achieves superior performance in zero-shot synthetic-to-real transfer (Table 2), indicating that our method learns structural cues that are more robust across domains than simple pixel-wise regression.
>
> Furthermore, the table below shows results of SHED against other methods that utilize structural cues, specifically those employing segmentation to enhance depth estimation. Simsar et al. [A] integrate over-segmentation to enforce object-level consistency, while Mousavian et al. [B] adopt a classical multi-task learning approach to refine depth using semantic boundaries.
>
> | Method            | $\delta > 1.25 $ $\uparrow$ | $\delta > 1.25^2$ $\uparrow$ |$\delta> 1.25^3$ $\uparrow$ | AbsRel $\downarrow$ | RMSE $\downarrow$ | log10 $\downarrow$ |
> |-------------------|------------|-------------|-------------|----------|--------|---------|
> | Simsar et al.     | 0.847      | 0.971       | 0.993       | 0.124    | 0.456  | –       |
> | Mousavian et al.  | 0.568      | 0.856       | 0.956       | 0.200    | 0.816  | 0.061   |
> | **SHED (ours)**   |**0.855**     |**0.974**     | **0.993**   | **0.123**  | **0.433**  | **0.052**   |
>
> [A] Object-aware Monocular Depth Prediction with Instance Convolutions (RA-L 22, Simsar, Enis, et al)
>
> [B] Joint semantic segmentation and depth estimation with deep convolutional networks (3DV 16, Mousavian et al)

---

> ### Author Response · Authors · 2025-11-21
>
> **[W6] Scalability**
>
> We clarify that our method does not use any segmentation ground-truth at any stage. The segment hierarchy in SHED is learned entirely from depth supervision, without leaking semantic labels or relying on external segmentation annotations. Since the segmentation structure emerges purely from the model’s inductive bias and optimization, SHED does not depend on the availability of segmentation GT and can be trained on large-scale unlabeled datasets in exactly the same way. Therefore, our approach naturally scales beyond NYUv2 and is not restricted by dataset size or segmentation label availability.

---

> > ### Comment · Reviewer_o4eZ · 2025-11-26
> > **Reply to Authors**
> >
> > Thanks the authors for providing these detailed replies. Most of my concerns are addressed.
> >
> > Currently, I respectfully have different opinions regarding [W1]:
> >
> > The authors claim: It is a novel architectural extension that transforms the encoder-only CAST framework, originally designed for recognition, into a symmetric encoder-decoder structure capable of dense prediction.
> >
> > My concern is that this extension from encoder to encoder-decoder structure is hard to be claimed as a major contribution. Back to 2018, [1] proposed to adopt imagenet pretrained encoder, originally designed for recognition, into a symmetric encnoder-decoder structure capable of dense prediction. This is a general implementation in depth feild. The major difference is that, the adopted network in SHED, becomes an additional extension. To be ensembled to depth estimation, a decoder has to be designed in this case. The decoder's design also follows the encoder's structure. It's hard for me to see a great contribution here.
> >
> > Also, it's hard to claim that SHED beats DAV2 in terms of robust zero-shot generalization. This paper only presents one Hypersim to NYUv2 benchmark. Both models are trained with a signle domain synthetic dataset, and this toy experiment cannot indicate much generalization ability.
> >
> > Given this, I keep my score unchanged now.
> >
> > [1] High Quality Monocular Depth Estimation via Transfer Learning (arXiv 2018)

---

> ### Author Response · Authors · 2025-11-26
>
> Thank you for your follow-up comment.
>
> The technical novelty we want to emphasize is not the presence of a decoder itself, but the object being decoded and the mechanism by which it is decoded. SHED decodes a learned segment hierarchy by explicitly inverting the grouping structure built in the encoder, rather than dense feature maps. In the encoder, SHED constructs a hierarchy of segment tokens via learned grouping over superpixels, resulting in a discrete tree-structured representation. In the decoder, we do not apply a generic CNN or ViT-style upsampling module. Instead, we perform coarse-to-fine unpooling using the stored assignment matrices, explicitly inverting the hierarchical grouping to reconstruct pixel-wise geometry. The reverse hierarchy therefore performs structure decoding by inverting segment-level relationships, rather than conventional feature decoding via interpolation.
>
> We agree that adopting an ImageNet-pretrained encoder and attaching a decoder for dense prediction, as done in prior work such as [1], is now standard. However, this framing does not characterize what SHED does. The encoder–decoder paradigm in [1] decodes dense feature tensors using convolutional upsampling and skip connections. SHED decodes a learned segment tree using hierarchical inversion. While both may be described at a very high level as encoder–decoder architectures, the similarity ends there. One reconstructs dense features, whereas the other reconstructs structure. What this structural decoding delivers — and existing approaches cannot — is cleaner and more coherent depth boundaries, features organized by global scene layout, and 3D reconstructions with emergent part structure beyond pixel-wise methods. In the broader view, SHED advances CAST by extending hierarchical reasoning from recognition into structure-aware generative reconstruction, an architectural capability beyond both CAST and conventional encoder–decoder designs.
>
> Regarding zero-shot generalization, we agree that a single benchmark should not be interpreted as a universal claim that SHED “outperforms DAV2.” We have provided zero-shot results from our mixed-domain training setup, showing consistent improvements over the DPT baseline across various datasets in [W2 & W3a] under matched training configurations. We also note that DPT and DepthAnything 2 share a largely similar encoder–decoder design.
>
> We also emphasize that SHED and DAV2 rely on fundamentally different generalization mechanisms. DAV2 is scale and data-driven, benefiting from massive pretraining and dataset diversity. SHED is structure-driven, relying on explicit hierarchical organization and its inversion during decoding. The HyperSim → NYUv2 benchmark was chosen as a concrete sim-to-real stress test with substantial domain shift. That SHED remains competitive in this setting despite using only ImageNet-1K pretraining indicates that geometry-centric inductive bias provides robustness that is not solely attributable to data scale.
>
> In summary, our core contribution is the decoding of a learned hierarchical structure for dense geometry reconstruction, which we believe is a clear architectural innovation beyond the standard encoder-decoder extension. Our generalization results simply emphasize the complementarity and robustness of this structural approach.

---

### Official Review · Reviewer_yMXn · 2025-10-31

**Soundness:** 2
**Presentation:** 2
**Contribution:** 2
**Rating:** 4
**Confidence:** 3

**Summary:**

The authors propose SHED, a novel encoder–decoder architecture that explicitly incorporates geometric priors from spatial layouts by integrating semantic segmentation into the depth estimation process.
Specifically, SHED transforms the input image into superpixel tokens and applies graph pooling to construct coarser segments during encoding, enhancing depth boundary sharpness, segment coherence, and cross-domain robustness.
Built upon the DPT backbone, SHED achieves superior performance on both in-domain and cross-domain depth estimation benchmarks compared to the original DPT model.

**Strengths:**

1. Incorporation of Superpixels into Depth Estimation
SHED integrates superpixel information into the ViT token representation within the depth estimation network, enabling the model to jointly exploit raw RGB features and segmentation-aware cues.
By embedding discrete object boundary information through superpixels, the network gains enhanced structural awareness, leading to improved scene layout understanding and depth prediction accuracy.

2. Performance on Depth Estimation and 3D Layout
The authors validate the effectiveness of SHED through both quantitative depth estimation metrics and qualitative 3D layout results.
Their experiments demonstrate that SHED not only achieves superior pixel-wise depth accuracy but also produces more coherent and geometrically consistent scene layouts.

**Weaknesses:**

1. Limited Novelty
The main contribution of this paper lies in the hierarchical clustering component, while the majority of the pipeline is directly adapted from existing frameworks such as DPT and CAST. As a result, much of the observed performance gain may stem from these strong baselines rather than the proposed SHED module itself. This concern is further supported by the narrow performance gap between DPT and SHED reported in the results table, which raises questions about the true contribution of the newly introduced components.

2. Absence of Ablation Study on Model Architecture
Although the proposed method claims architectural innovation, the paper lacks an ablation study to validate the contribution of each architectural component. Without such analysis, it is difficult to determine whether the performance improvement originates from the encoder, decoder, or hierarchical clustering mechanism, and to what extent each contributes to the overall gain.

3. Missing Evaluation on Zero-Shot Depth Estimation
As far as I understand, the cross-domain evaluation in this paper still involves shared datasets for fine-tuning and testing, which limits its generalization claims. Given that recent trends in depth estimation research emphasize zero-shot performance across unseen domains, the absence of such evaluation weakens the paper’s claim of robust cross-domain generalization.

4. Insufficient Comparison with State-of-the-Art Methods
The experimental comparisons are primarily conducted against DPT, which is no longer representative of the current state-of-the-art. To strengthen the validity and competitiveness of SHED, the authors should include comparisons with more recent high-performing models such as Marigold and DepthPro. This would provide a fairer and more persuasive assessment of the proposed method’s effectiveness.

**Questions:**

1. What happened if we drop hierarchical clustering with the same SHED decoder or keep hierarchical clustering with DPT decoder? The paper needs some ablation study.

2. What is the performance of SHED with zero-shot depth estimation task (evaluation on NYUv2, KITTI, ETH3D, ScanNet, DIODE)?

3. Please refer to the upper weakness part

---

> ### Author Response · Authors · 2025-11-21
>
> Dear reviewer yMXn,
>
> Thank you for recognizing the importance of incorporating structural cues into depth estimation and for highlighting the effectiveness of our approach. We appreciate your positive remarks regarding the enhanced structural awareness enabled by our segment hierarchy, as well as the strong results demonstrated in both depth estimation and 3D layout representations. We address your concerns and questions in detail below.
>
> **[W1] Novelty and performance gains**
>
> We clarify that SHED is not a mere combination of existing modules but a novel architectural extension that transforms the encoder-only CAST framework, originally designed for recognition, into a symmetric encoder-decoder structure capable of dense prediction. The core novelty lies in our proposed reverse hierarchy decoder, which uniquely unpools abstract segment tokens to reconstruct pixel-wise geometry, a mechanism absent in prior work. Regarding performance, since we strictly controlled the backbone for a fair comparison, the observed gains are entirely attributable to our method. Furthermore, SHED demonstrates significant structural superiority, evidenced by sharper boundaries, robust zero-shot generalization outperforming Depth Anything v2, and a 34% improvement of top-1 recall in layout-aware scene retrieval.
>
> **[W2 & Q1] Absence of ablation study on model architecture**
>
> We appreciate the reviewer’s constructive feedback regarding the verification of individual architectural components. To address this, we conducted an additional ablation study to isolate the contributions of the hierarchical clustering in the encoder and the hierarchy reversing in the decoder. To verify the necessity of our proposed decoder, we experimented with a variant of SHED by removing the progressive unpooling process in the decoder. The quantitative results on the NYUv2 dataset are presented below.
>
> \begin{array}{lcccccc}
> \text{Method}  & \delta > 1.25 \uparrow  & \delta >  1.25^2 \uparrow &  \delta >  1.25^3 \uparrow &\text{AbsRel} \downarrow &\text{RMSE} \downarrow & \log10 \downarrow  \newline
> \hline
> \text{DPT} & 0.839 & 0.971  &  \textbf{0.992}
>                                  &  0.132 & 0.457 &  0.055 \newline
> \text{SHED (ours) with forward hierarchy only} & 0.755 & 0.951 & 0.989 & 0.163 & 0.552 &  0.069  \newline
> \text{SHED (ours)} & \textbf{0.846} & \textbf{0.972} & \textbf{0.992} & \textbf{0.130} & \textbf{0.451} & \textbf{0.054}
> \newline
> \end{array}
>
> As shown in the table, removing the reverse hierarchy leads to a sharp performance degradation, falling significantly behind the DPT baseline. This demonstrates that the coarse-to-fine unpooling mechanism in the decoder is essential to recover fine-grained spatial details from the grouped representations. Conversely, attaching our SHED decoder to a standard ViT is architecturally infeasible. The decoder’s unpooling operation mathematically relies on the specific assignment matrices ($P_l$) generated during the encoder’s grouping phase. Without this specific grouping history, the reverse hierarchy cannot be constructed.

---

> ### Author Response · Authors · 2025-11-21
>
> **[W3 & Q2] Evaluation on diverse benchmarks**
>
> We extend our experiments to a mixed-domain training setup using Virtual KITTI, HyperSim, and MegaDepth [1]. We train the SHED using a uniform sampling strategy across all datasets for 20 epochs with batch size of 64. We then evaluate this model in a fully zero-shot manner on four unseen datasets, including KITTI (outdoor), NYUv2 (indoor), SUN-RGBD (indoor).
>
> The results below demonstrate that SHED consistently outperforms the DPT baseline across domains, including both indoor and outdoor scenes. The segment-hierarchy inductive bias of SHED is not specific to the NYUv2 domain but transfers effectively to outdoor and mixed-domain settings, consistently outperforming the baseline.
>
> | Method        | KITTI AbsRel $\downarrow$ | KITTI $\delta$>1.25 $\uparrow$ | NYUv2 AbsRel $\downarrow$  | NYUv2 $\delta$>1.25 $\uparrow$ | SUN-RGBD AbsRel $\downarrow$  | SUN-RGBD $\delta$>1.25 $\uparrow$  |
> |---------------|-----------------|-----------------|-----------------|-----------------|--------------------|--------------------|
> | DPT           | 0.286               | 0.475               | 0.247               | 0.559              |           8.58        | 0.169                  |
> | SHED (ours)  | **0.272**              | **0.500**           |  **0.244**              | **0.571**               | **7.16**                  | **0.190**                  |
>
> [1] Megadepth: Learning single-view depth prediction from internet photos (CVPR 18, Li, Zhengqi)
>
>
> **[W4] Comparison with state-of-the-art models**
>
> Thank you for this suggestion. To address the question regarding competitiveness with the latest depth models, we have added a direct comparison with Marigold, where the model is pre-trained with over five billion images. We use their publicly released checkpoints and evaluation protocols. Since Marigold predicts relative depth, we applied scale and shift alignment using the ground truth following standard relative depth evaluation protocols. As shown in the table below, SHED shows highly competitive performance compared to both Depth Anything v2 and Marigold in cross-domain (synthetic $\rightarrow$ real) evaluation from HyperSim to NYUv2, despite using more constrained priors, ImageNet-1K. This demonstrates that the structural inductive bias of SHED yields superior generalization in the sim-to-real gap, even when compared to foundation models pre-trained on vastly larger datasets.
>
> | Method              | Pre-training | Training  | δ > 1.25 ↑ | δ > 1.25² ↑ | δ > 1.25³ ↑ | AbsRel ↓     | RMSE ↓      | log10 ↓    |
> |---------------------|--------------|-----------|------------|-------------|-------------|--------------|-------------|------------|
> | **Marigold**        | Laion 5B     | HyperSim  | 0.375      | 0.659       | 0.833       | **0.542**    | 1.243       | 0.171      |
> | **Depth Anything 2**| LVM-142M     | HyperSim  | 0.592      | **0.902**   | **0.960**   | 0.749        | 0.808       | 0.110      |
> | **SHED (ours)**     | IN-1K        | HyperSim  | **0.632**  | 0.892       | **0.960**   | 0.583        | **0.740**   | **0.102**  |
>
> Regarding DepthPro, its main advantages come from focal-length conditioning and training on extremely large datasets. Our contribution, injecting a hierarchical structural bias into the ViT encoder, is orthogonal to these design choices. Since DepthPro relies on a standard ViT backbone, the SHED hierarchy could be integrated into such models to further enhance performance. We consider this a promising direction for future work.

---

### Official Review · Reviewer_o4PH · 2025-11-01

**Soundness:** 3
**Presentation:** 3
**Contribution:** 2
**Rating:** 4
**Confidence:** 4

**Summary:**

This paper proposes SHED, a ViT-based model for monocular depth estimation. The method introduces a bidirectional segment hierarchy into the DPT framework. On the encoder side, the model follows the CAST strategy by replacing traditional square patch tokens with superpixel tokens, which are progressively clustered by feature similarity to form a fine-to-coarse hierarchical representation. On the decoder side, a reverse hierarchy is introduced, performing soft unpooling and segment-to-pixel projection to remap hierarchical semantic features back to spatial feature maps, thereby generating structured depth outputs. The design treats segmentation as a geometric structural prior, enhancing boundary sharpness and intra-segment consistency in depth prediction. Compared with DPT-based baselines, SHED achieves improvements in object boundary quality, local depth consistency, and layout-aware retrieval performance.

**Strengths:**

- Clear presentation: concepts (superpixel/segment tokens, soft unpooling) and implementation details are well explained and reproducible.
- Strong results: sharper boundaries, better intra-segment consistency, and improved layout-aware retrieval over DPT-based baselines.

**Weaknesses:**

1. Limited in novelty.
   While the bidirectional segment hierarchy is somewhat new, the broader idea of leveraging segmentation structure to assist depth estimation has been explored [1]. The paper should more clearly delineate how SHED differs from prior joint semantic–segmentation–depth approaches and what new contributions it adds.
2. Insufficient experiments.
   The paper lacks a quantitative evaluation of the learned segmentation results and only presents visualizations. It should add comparisons of segmentation metrics such as F1 score and IoU, similar to CAST. While the paper shows better results than DepthAnything in the constrained, fine-tuned setting, it's crucial to show DepthAnything's zero-shot performance on NYU as well. The inferior performance of DA may be caused by the fine-tuning process instead of the model itself.
3. Lack of efficiency analysis.
   Although the method emphasizes structural advantages, it provides no quantitative comparison against DPT baselines in terms of FLOPs, parameter count, inference speed, or GPU memory.
4. Lack of comparison with segmentation-guided depth estimation methods.
   The paper’s main claim is that segmentation structures help with depth estimation, but the comparisons do not include some state-of-the-art methods that also utilize segmentation to improve depth estimation.

--
[1] Object-aware Monocular Depth Prediction with Instance Convolutions.

**Questions:**

1. The authors present boundary results for SHED and DPT in Figure 4. Although SHED has more complete boundaries than DPT, it appears that SHED introduces more erroneous boundaries. Could the authors explain this?
2. The experiments mention only the NYUv2 dataset. Could you provide results on an additional dataset such as KITTI?

---

> ### Author Response · Authors · 2025-11-21
>
> Dear reviewer o4PH,
>
> We appreciate your positive assessment of our clear explanation of concepts such as segment tokens and soft unpooling, as well as the reproducibility of our implementation. We are also grateful for highlighting our strong results, including sharper boundaries, improved intra-segment consistency, and enhanced layout-aware retrieval. We address your concerns and questions in detail below.
>
> **[W1] Separation of contributions from joint semantic segmentation-depth  approaches**
>
> SHED differs from existing segmentation-assisted depth estimation approaches in several fundamental ways.
>
> SHED adopts a fundamentally different perspective on segmentation. In our work, segmentation is not an external auxiliary signal provided by off-the-shelf models or human annotations. Instead, it is a learned structural representation that emerges entirely from depth supervision. SHED discovers a bidirectional segment hierarchy internally without any explicit segmentation labels.
>
> In particular, the resulting segments differ qualitatively from semantic segmentation as our hierarchy is learned via depth reconstruction. As shown in Fig. 6, SHED decomposes large surfaces (e.g., a floor) based on depth variations and groups spatially consistent regions. This reveals a geometry-aware organization that standard appearance-based or semantic segmentation methods fail to capture.
>
> Our results also demonstrate that this learned hierarchy significantly improves cross-domain generalization. Compared to DPT and Depth Anything V2, SHED achieves superior performance in zero-shot synthetic-to-real transfer (Table 2), indicating that our method learns structural cues that are more robust across domains than simple pixel-wise regression.

---

> ### Author Response · Authors · 2025-11-21
>
> **[W2a] Benchmarking semantic segmentation**
>
> We evaluate the quality of segmentation using semantic segmentation metrics on the ADE20K dataset after fine-tuning. Following CAST evaluation protocol, we compute boundary F-score and region IoU between the predicted segments and ground-truth segmentation mask. As shown in Table, \sname achieves comparable or higher alignment with ground-truth semantic structures. This highlights that \sname not only leverages the hierarchy effectively for depth prediction but also learns segment boundaries that correspond well to meaningful scene geometry.
>
> | Method        | mIoU | Boundary F-score |
> |---------------|------|------------------|
> | CAST          | 43.1 | 36.5             |
> | SHED (ours)   | 44.5 | 37.7             |
>
> **[W2b] Zero-shot performance**
>
> Regarding the comparison with Depth Anything 2 (DA2), we would like to clarify that we utilized the official fine-tuning codebase and checkpoints provided by the DA2 authors.
>
> It is important to note that the zero-shot DA2 is a relative depth model, whereas our fine-tuning experiments evaluate metric depth estimation. To evaluate DA2 on metric benchmarks like NYUv2, we applied scale and shift alignment using the ground truth following standard relative depth evaluation protocols. This process effectively provides the zero-shot model with oracle scale information, resulting in high metrics.
>
> In contrast, the metric fine-tuning experiments on HyperSim requires the model to predict absolute depth values without access to ground-truth statistics at test time. Under this setting, while DA2 suffers from a domain gap, SHED consistently outperforms the fine-tuned DA2.
>
> | Method                         | Pre-training | Training  | GT alignment | δ > 1.25 ↑ | δ > 1.25² ↑ | δ > 1.25³ ↑ | AbsRel ↓ | RMSE ↓ | log10 ↓ |
> |--------------------------------|--------------|-----------|--------------|------------|-------------|-------------|----------|--------|---------|
> | **Depth Anything 2 (zero-shot)** | LVM-142M     | -         | ✓            | 0.835      | 0.966       | 0.985       | 0.143    | 0.876  | 0.058   |
>
> ---
>
> | Method            | Pre-training | Training | GT alignment | δ > 1.25 ↑ | δ > 1.25² ↑ | δ > 1.25³ ↑ | AbsRel ↓ | RMSE ↓ | log10 ↓ |
> |-------------------|--------------|----------|--------------|------------|-------------|-------------|----------|--------|---------|
> | **Depth Anything 2** | LVM-142M   | HyperSim | -            | 0.592      | **0.902**   | **0.960**   | 0.749    | 0.808  | 0.110   |
> | **SHED (ours)**     | IN-1K      | HyperSim | -            | **0.632**  | 0.892       | **0.960**   | **0.583**| **0.740** | **0.102** |

---

> ### Author Response · Authors · 2025-11-21
>
> **[W3] Efficiency analysis**
>
>
> We report parameters and FLOPs for a fair, hardware-independent comparison of structural complexity. As shown in the table below, SHED reduces the computational cost GFLOPs by approximately 24% compared to DPT-Hybrid, indicating superior structural efficiency and lower theoretical latency.
>
> | Method | Params (M) | FLOPs (G) |
> | -----| ----- | ----- |
> | DPT-Hybrid | 41.88 | 135.0 |
> | SHED (ours) | 56.58 | 103.2 |

---

> ### Author Response · Authors · 2025-11-21
>
> **[W4] Comparison with segmentation-guided depth estimation methods**
>
> Table shows comparison results of SHED against other methods that utilize structural cues, specifically those employing segmentation to enhance depth estimation. Simsar et al. [A] integrate over-segmentation to enforce object-level consistency, while Mousavian et al. [B] adopt a classical multi-task learning approach to refine depth using semantic boundaries.
>
> | Method            | $\delta > 1.25 $ $\uparrow$ | $\delta > 1.25^2$ $\uparrow$ |$\delta> 1.25^3$ $\uparrow$ | AbsRel $\downarrow$ | RMSE $\downarrow$ | log10 $\downarrow$ |
> |-------------------|------------|-------------|-------------|----------|--------|---------|
> | Simsar et al.     | 0.847      | 0.971       | 0.993       | 0.124    | 0.456  | –       |
> | Mousavian et al.  | 0.568      | 0.856       | 0.956       | 0.200    | 0.816  | 0.061   |
> | SHED (ours)  | **0.855**     | **0.974**     |**0.993**   | **0.123**  |**0.433**  | **0.052**   |
>
> [A] Object-aware Monocular Depth Prediction with Instance Convolutions (RA-L 22, Simsar, Enis, et al)
>
> [B] Joint semantic segmentation and depth estimation with deep convolutional networks (3DV 16, Mousavian et al)

---

> ### Author Response · Authors · 2025-11-21
>
> **[Q1] Clarification of visualization**
>
> While SHED may display additional boundary activations, these are not erroneous edges but a natural consequence of its higher geometric sensitivity. As noted in Section 4.2, DPT often misses subtle depth discontinuities, resulting in low recall and blurry shapes. In contrast, SHED's segment-based approach ensures sharper transitions and captures fine details that DPT ignores. Consequently, SHED achieves significantly better quantitative performance as shown in Table 1.
>
> **[Q2] Evaluation on diverse benchmarks**
>
> we extend our experiments to a mixed-domain training setup using Virtual KITTI, HyperSim, and MegaDepth [1]. We train the SHED using a uniform sampling strategy across all datasets for 20 epochs with batch size of 64. We then evaluate this model in a fully zero-shot manner on four unseen datasets, including KITTI (outdoor), NYUv2 (indoor), SUN-RGBD (indoor).
>
> The results below demonstrate that SHED consistently outperforms the DPT baseline across domains, including both indoor and outdoor scenes. The segment-hierarchy inductive bias of SHED is not specific to the NYUv2 domain but transfers effectively to outdoor and mixed-domain settings, consistently outperforming the baseline.
>
> | Method        | KITTI AbsRel $\downarrow$ | KITTI $\delta$>1.25 $\uparrow$ | NYUv2 AbsRel $\downarrow$  | NYUv2 $\delta$>1.25 $\uparrow$ | SUN-RGBD AbsRel $\downarrow$  | SUN-RGBD $\delta$>1.25 $\uparrow$  |
> |---------------|-----------------|-----------------|-----------------|-----------------|--------------------|--------------------|
> | DPT           | 0.286               | 0.475               | 0.247               | 0.559              |           8.58        | 0.169                  |
> | SHED (ours)  | **0.272**   |**0.500**           |  **0.244**              |**0.571**               | **7.16**                  | **0.190**                  |
>
>
> [1] Megadepth: Learning single-view depth prediction from internet photos (CVPR 18, Li, Zhengqi)

---

### Official Review · Reviewer_UhUV · 2025-11-01

**Soundness:** 3
**Presentation:** 4
**Contribution:** 3
**Rating:** 6
**Confidence:** 5

**Summary:**

This paper presents a superpixel-based method for dense prediction via hierarchical grouping. The method, named SHED, operates on the internal feature space and creates a hierarchy of superpixel-level segments across which both a fine-to-coarse aggregation and a coarse-to-fine refinement are performed. The primary task of interest on which the method is validated is monocular depth estimation / 3D estimation, which is shown to benefit from the segment-level grouping when working with a modern DPT-based network. The authors also demonstrate quantitative improvements in other related tasks over the baseline network, such as boundary detection and image retrieval.

**Strengths:**

1. The paper is well-written and easy to follow. The method section is well-structured and the mathematical notation is defined rigorously.

2. The idea of applying a reverse segmentation hierarchy for coarse-to-fine refinement in a dense prediction task such as depth estimation is novel, interesting, well-motivated, and shown to yield promising results.

**Weaknesses:**

1. The method is built on a modern yet not state-of-the-art network for depth estimation, i.e. DPT, which dates from 2021. Apart from the comparison to Depth Anything v2 in Table 2, the authors have not considered any other recent and high-performing ViT-based networks to show their improvement holds in a more competitive setting, e.g. [A], [B], [C].

2. For the depth estimation training and evaluation, the authors have trained SHED only on two datasets (separately on each), i.e. NYUv2 and HyperSim, and used the test set of only one of these two datasets (NYUv2) to evaluate their method. This practice has two issues. First, the training data are not diverse, while recent methods [A], [B], [C] have shown that leveraging large-scale, diverse training data is equally important to a sophisticated model design for well-generalizing depth estimation. Second, the limited-domain evaluation which is performed by the authors is less indicative of the true ability of the trained model to generalize, which is why most recent works primarily focus on the zero-shot evaluation setting, testing a single universally trained model on *several different datasets* from those included in training. It would have been beneficial to perform such diverse zero-shot evaluation for SHED on depth estimation too.

[A] UniDepth: universal monocular metric depth estimation. In CVPR, 2024.

[B] Metric3D: towards zero-shot metric 3D prediction from a single image. In ICCV, 2023.

[C] UniK3D: universal camera monocular 3D estimation. In CVPR, 2025.

**Questions:**

1. Can the authors include more methods in a fair, in-domain comparison for depth estimation with training and testing on NYUv2, beyond just DPT and Depth Anything v2?

2. Can the authors extend their evaluation to a more complete and diverse set of evaluation datasets?

---

> ### Author Response · Authors · 2025-11-21
>
> Dear reviewer UhUV,
>
> Thank you for your thoughtful and positive assessment of our work. We sincerely appreciate your comments highlighting that the paper is well-written and easy to follow, that our method section is well-structured with rigorous mathematical notation, and that the idea of using a reverse segmentation hierarchy for coarse-to-fine refinement in dense prediction is novel, interesting, and well-motivated. We address your concerns and questions in detail below.
>
> **[W1 & Q1] Comparison with state-of-the-art models**
>
> To address the question regarding competitiveness with the latest depth models, we have added a direct comparison with Marigold, where the model is pre-trained with over five billion images. We use their publicly released checkpoints and evaluation protocols. Since Marigold predicts relative depth, we applied scale and shift alignment using the ground truth following standard relative depth evaluation protocols. As shown in the table below, SHED shows highly competitive performance compared to both Depth Anything v2 and Marigold in cross-domain (synthetic $\rightarrow$ real) evaluation from HyperSim to NYUv2, despite using more constrained priors, ImageNet-1K. This demonstrates that the structural inductive bias of SHED yields superior generalization in the sim-to-real gap, even when compared to foundation models pre-trained on vastly larger datasets.
>
> \begin{array}{l|lcccccc}
> \text{Method}  & \text{Pre-training} & \text{Training} & \delta > 1.25 \uparrow  & \delta >  1.25^2 \uparrow &  \delta >  1.25^3 \uparrow &\text{AbsRel} \downarrow &\text{RMSE} \downarrow & \log10 \downarrow  \newline
> \hline
> \text{Marigold} & \text{Laion 5b} & \text{HyperSim} & 0.375 & 0.659  &  0.833
>                                  &  \textbf{0.542} & 1.243 &  0.171 \newline
> \text{Depth Anything 2} & \text{LVM-142M} & \text{HyperSim} & 0.592 & \textbf{0.902} & \textbf{0.960} & 0.749 & 0.808 &  0.110  \newline
> \text{SHED (ours)}  & \text{IN-1K} & \text{HyperSim} & \textbf{0.632} & 0.892 & \textbf{0.960} & 0.583 & \textbf{0.740} & \textbf{0.102}
> \newline
> \end{array}
>
> We select DPT as an established baseline to isolate the effectiveness of our segment hierarchy contribution in depth estimation. Regarding recent methods like UniDepth [A], Metric3D [B], and UniK3D [C], their primary gains come from bootstrapping dense camera predictions and training on massive datasets. Our contribution, incorporating hierarchical structure into the ViT encoder, is orthogonal and complementary to these strategies. Since these foundation models rely on standard ViT backbones, the SHED design principle could theoretically be integrated into them to further boost performance. We leave this integration in future work.
>
> **[W2 & Q2] Evaluation on diverse benchmarks**
>
> To address the concern regarding limited domain evaluation, we extend our experiments to a mixed-domain training setup using Virtual KITTI, HyperSim, and MegaDepth [1]. We train the SHED using a uniform sampling strategy across all datasets for 20 epochs with batch size of 64. We then evaluate this model in a fully zero-shot manner on four unseen datasets, including KITTI (outdoor), NYUv2 (indoor), SUN-RGBD (indoor).
>
> The results below demonstrate that SHED consistently outperforms the DPT baseline across domains, including both indoor and outdoor scenes. The segment-hierarchy inductive bias of SHED is not specific to the NYUv2 domain but transfers effectively to outdoor and mixed-domain settings, consistently outperforming the baseline.
>
> | Method        | KITTI AbsRel $\downarrow$ | KITTI $\delta$>1.25 $\uparrow$ | NYUv2 AbsRel $\downarrow$  | NYUv2 $\delta$>1.25 $\uparrow$ | SUN-RGBD AbsRel $\downarrow$  | SUN-RGBD $\delta$>1.25 $\uparrow$  |
> |---------------|-----------------|-----------------|-----------------|-----------------|--------------------|--------------------|
> | DPT           | 0.286               | 0.475               | 0.247               | 0.559              |           8.58        | 0.169                  |
> | SHED (ours)  | **0.272**   |**0.500**           |  **0.244**              |**0.571**               | **7.16**                  | **0.190**                  |
>
> [1] Megadepth: Learning single-view depth prediction from internet photos (CVPR 18, Li, Zhengqi)

---

### Author Response · Authors · 2025-11-29

Dear Area Chair,

We sincerely appreciate your time and effort in serving the community. We first summarize the positive aspects highlighted by each reviewer.

- Reviewer UhUV: (1) The paper is well-written and easy to follow, with a well-structured method section and rigorous mathematical notation. (2) The idea of applying a reverse segmentation hierarchy for coarse-to-fine refinement in dense prediction is novel, interesting, and well-motivated.

- Reviewer o4PH: (1) Clear presentation of concepts such as superpixel/segment tokens and soft unpooling, with reproducible implementation details. (2) Strong improvements in boundary sharpness, intra-segment consistency, and layout-aware retrieval over DPT-based baselines.

- Reviewer yMXn: (1) Incorporation of superpixel/structural cues into depth estimation, enabling enhanced structural awareness and scene layout understanding. (2) Strong performance on both depth metrics and 3D layout, producing more coherent and geometrically consistent scene structures than the DPT baseline.

- Reviewer o4eZ:  (1) The paper is well-motivated; introducing a segmentation process into monocular depth estimation is reasonable and natural. (2) Layout-aware retrieval is an interesting and appropriate metric to assess the gain from CAST-style structure. (3) Large gains in boundary and part-level structure empirically demonstrate the effectiveness of SHED.

In this rebuttal, we have shown that:

1. Clarifying technical novelty: structure decoding vs. feature decoding
- We clarified that SHED **decodes a learned segment hierarchy** by explicitly inverting the grouping structure built in the encoder, rather than dense feature maps.
- SHED constructs, in the encoder, a tree-structured segment hierarchy through learned grouping over superpixels.
- In the decoder, SHED does not perform generic CNN or ViT upsampling of dense feature maps. Instead, it performs **coarse-to-fine unpooling using stored assignment matrices**, explicitly inverting the grouping structure.
- We explain that this structure decoding yields **cleaner and more coherent depth boundaries, features organized by global scene layout, and 3D reconstructions with emergent part structure**.
- We also clarify that SHED **never uses segmentation ground truth** at any stage and The segment hierarchy emerges purely from depth supervision.

2. Ablation on architectural components
- We added an ablation study that removes the reverse hierarchy in the decoder of SHED.
- The ablation shows a sharp performance degradation, demonstrating that the hierarchy inversion is essential for recovering fine-grained spatial details from grouped representations.
- We also clarify why attaching the SHED decoder to a standard ViT/DPT encoder is architecturally infeasible. The decoder’s unpooling mathematically relies on the assignment matrices produced by the encoder’s grouping steps. Without that grouping history, the reverse hierarchy cannot be defined.

3. Broader and more diverse zero-shot / multi-domain evaluation
- We introduced mixed-domain training on Virtual KITTI, HyperSim, and MegaDepth.
- We then evaluate zero-shot on unseen configurations, including: KITTI, NYUv2, SUN-RGBD.
- Across these datasets, SHED consistently outperforms the DPT baseline under matched training configurations, showing that the segment-hierarchy inductive bias is not specific to NYUv2, and transfers to both indoor and outdoor domains under a more scalable, diverse training setup.

4. Comparisons with stronger state-of-the-art depth models
- We added direct comparisons with Marigold and Depth Anything v2 (DAV2-small and DAV2-base).
- Marigold is pre-trained on Laion-5B images, and DAV2 on LVM-142M; SHED uses only ImageNet-1K.
- Under HyperSim training and HyperSim→NYUv2 transfer, SHED achieves competitive or superior cross-domain performance compared to both DAV2 and Marigold, despite the much more limited pretraining data.

5. Quantitative evaluation of the learned segmentation
- We added semantic segmentation metrics on ADE20K following the CAST protocol.
- SHED achieves comparable or higher alignment with semantic structures than CAST.

6. Comparison with segmentation-guided depth baselines
- We added comparisons with segmentation-guided depth methods that leverage structural cues.
- SHED surpasses these methods without using semantic labels.

7. Efficiency, FLOPs, and scalability
- SHED reduces FLOPs by approximately 24% compared to DPT-Hybrid while improving accuracy, indicating lower latency.
- DAV2-base exhibits the highest FLOPs among all methods, while SHED operates with substantially lower FLOPs and still achieves competitive cross-domain performance.

We believe these revisions address the main concerns regarding novelty, evaluation breadth, robustness, and efficiency, and clarify SHED’s contribution as an architectural framework for decoding a learned segment hierarchy for dense geometry reconstruction.

Sincerely,

Authors.

---

### Meta-Review · Area_Chair_owiq · 2026-01-07

**Summary:**

The paper presents a method to enhance monocular depth estimation with segmentation. It received mixed ratings from four reviewers. Most of the reviewers provided negative ratings and raised critical concerns regarding the paper's key contribution and experimental evaluation. Their concerns mainly include the incremental novelty of the technical contribution (the main motivation, i.e., segmentation for depth estimation, has been widely investigated in prior works), the insufficient experimental comparison with advanced state-of-the-art approaches and the biased evaluation data distribution, and the missing important ablation studies to show the effectiveness and the verification of the model efficiency perspective. Based on the overall ratings and the comments, AC finally decided to recommend a rejection of this submission this time.

**Reviewer Concerns:**

The major concerns from the reviewers include the following aspects: (i) the incremental novelty of the technical contribution (the main motivation, i.e., segmentation for depth estimation, has been widely investigated in prior works),  (ii) the insufficient experimental comparison with advanced state-of-the-art approaches and the biased evaluation data distribution, and (iii) the missing important ablation studies to show the effectiveness and the verification of the model efficiency perspective. The rebuttal can only partially address these critical issues. Large modifications are necessary.

**Reviewer Scores:**

The original review score distribution is 6, 4, 4, 4. The reviewers did not indicate any intention to increase the scores after the rebuttal. So the final decision is straightforward.

---

### Decision · Program_Chairs · 2026-01-26

Reject